# A nanotwinned-alloy strategy enables fast sodium deposition dynamics

Guodong Zou[1], Jinming Wang [1,2] ✉, Yong Sun[1], Weihao Yang[1], Tingting Niu[1], Jinyu Li[1], Liqun Ren[3], Zhi Wei Seh [2] & Qiuming Peng [1] ✉

Sodium (Na) metal batteries are considered promising solutions for next-generation electrochemical energy storage because of their low costs and high energy densities. However, the slow Na dynamics result in unfavorable Na deposition and dendrite growth, which compromise cycling performance. Here we propose a nanotwinned alloy strategy prepared by high-pressure solid solution followed by Joule-heating treatment to address sluggish Na dynamics, achieving homogeneous Na deposition. By employing cost-effective Al-Si alloys for validation, Si solubility of 10 wt.% is extended through a high-pressure solid solution, and nanotwinned-Si particles, with a volume fraction of 82.7%, are subsequently formed through Joule-heating treatment. The sodiophilic nanotwinned-Si sites exhibit a high diffusion rate, which reduces the nondimensional electrochemical Damköhler number to far below 1, shifting the diffusion-controlled deposition behavior to reaction-controlled deposition. This transition facilitates spherical Na deposition and dendrite-free growth, allowing a symmetric cell to achieve stable Na plating/stripping over 5300 h at 5 mA cm$^{-2}$ with a cumulative capacity of 13.25 Ah cm$^{-2}$. This strategy is also demonstrated in another CuAg system with nanotwinned Ag structures.

The rapid growth and adoption of electrochemical energy storage in our society has pushed everyone to develop next-generation batteries with high capacities, long-term operation times and low costs[1–3]. Among the candidates, Na metal batteries (NMBs) have emerged as the most promising alternatives for post-Li-ion technology. They have three advantages. (i) Na batteries leverage identical operating principles and structural designs as their Li-based counterparts, thereby enabling the utilization of existing manufacturing infrastructure that was originally developed for Li batteries[4]. (ii) Cost-competitive Na is highly abundant in the Earth's crust, with a concentration of $23.6 \times 10^3$ ppm, surpassing the quantity of Li by three orders of magnitude (20 ppm)[5]. (iii) Na ions cannot be alloyed with Al; thus, relatively inexpensive Al can be used as a collector for both the positive and negative electrodes of Na batteries. Specifically, the theoretical replacement of Cu with Al and of Li with Na results in a reduction of total

cost, where the implied cost of Cu-Al replacement is 2.3 times greater than that of Li-Na replacement[2].

Na metal has emerged as the optimal negative electrode for Na batteries due to its low redox potential (−2.714 V vs. standard hydrogen electrode) and impressive specific capacity (1166 mAh g$^{-1}$)[4]. However, the slow Na dynamics leading to the occurrence of dendritic and nonplanar Na morphologies during electrodeposition hinders its reversibility, resulting in electrochemically active Na negative electrodes being underutilized[6,7]. Surface modifications of Na metal have been extensively studied, and they are shown to be effective at ensuring uniform Na deposition, whereas anodic modifications often lead to substantial safety concerns due to the high reactivity of Na[8]. Evidently, the morphology of deposited Na is significantly influenced by the choice of current collectors, especially at substantial discharge depths[9]. Consequently, significant efforts have

[1]State Key Laboratory of Metastable Materials Science and Technology, Yanshan University, Qinhuangdao, PR China. [2]Institute of Materials Research and Engineering (IMRE), Agency for Science, Technology and Research (A*STAR), 2 Fusionopolis Way, Innovis #08-03, Singapore 138634, Republic of Singapore. [3]Laboratory of Spinal Cord Injury and Rehabilitation, Chengde Medical University, Chengde, PR China. ✉e-mail: wangjinming1996@gmail.com; pengqiuming@ysu.edu.cn

been dedicated to developing cutting-edge collectors in the last decade, primarily including the creation of sodiophilic coatings to enhance dynamics and three-dimensional (3D) porous skeletons as substrates for Na deposition[10–12]. These materials are implemented to alleviate inhomogeneous Na deposition and Na dendrite growth. Nevertheless, the coatings are prone to separating from the collectors and swelling[13]. Furthermore, the use of a 3D host collector can lead to inefficient utilization of internal space[14]. Therefore, it is a great challenge to design a collector enhancing Na dynamics for guiding homogeneous Na nucleation and promoting dense planar Na growth.

Interestingly, alloying has emerged as a proficient method for modifying physicochemical properties[15]. Modulating chemical compositions and thermodynamic parameters, such as composition and temperature, can significantly influence alloy structures (grain boundaries and intermetallic compounds) and has been applied to various metal batteries[15–17]. Generally, the solidification of eutectic structures, with their strong adhesion characteristics, can effectively enhance the interfacial stability of a material. Nevertheless, owing to the large size, slow diffusion rate, and limited surface coverage characteristics of nucleation cores, active metals are prone to being deposited on the alloy matrix, inevitably resulting in dendrite growth. Theoretically, twin defects, which are mirror planes that divide a crystal lattice into two symmetric parts, are considered highly stable defects in alloy. Highly dense twin defects not only endow materials with extremely high electrical conductivities and mechanical strengths but can also effectively reduce the energy barrier to diffusion, thus enhancing the diffusion dynamics[18,19]. However, the current solid-phase-based methods for preparing numerous twin defects in alloys are limited by slow reaction rates and low volume fractions[16]. Notably, pressure, which is a thermodynamic parameter analogous to temperature and composition, can enhance solute solid solubility at high temperatures; then, high-temperature melt metastable states can be captured in terms of high degrees of supercooling for instantaneously unload pressure[20,21]. In addition, it has been reported that the Joule-heating (JH) process acts as an additional driving force by altering the free energy of the alloy to induce nonequilibrium relaxation, resulting in the formation of many crystal defects[22]. Therefore, it is anticipated that the high-pressure solid solution combined with Joule-heating ageing (HPJH) strategy might be a possible method for obtaining unique alloy collectors.

Here, we report a nanotwinned alloy collector strategy by HPJH to overcome the challenges of Na dendrite growth due to slow Na dynamics. As a proof of concept, we have chosen AlSi alloy collectors for this study. First, the solid solubility of Si in Al increases from 1.6 at.% in the atmosphere to ~10 at.% at 5 GPa[23]. Therefore, during the subsequent JH aging process, the occurrence of nonequilibrium relaxation leads to the preferential precipitation of nanotwinned Si (NT-Si) particles. Second, Si does not form an alloy with Na, which can prevent volume expansion during Na plating/stripping and reduce the risk of collector crushing[24]. Finally, AlSi alloys are cost-effective raw materials with a cost advantage during scaling. Experimental and theoretical results show that sodiophilic NT-Si exhibits higher diffusion rate, which reduces the nondimensional electrochemical Damköhler number and transitions the deposition process from diffusion-controlled to reaction-controlled. This transition promotes spherical Na deposition and dendrite-free growth. Consequently, a HPJH-AlSi alloy collector can deliver impressive long-term plating/stripping stability (exhibiting stability for 5300 h at 5 mA cm$^{-2}$ under 5 mA h cm$^{-2}$ and a cumulative capacity of 13.25 Ah cm$^{-2}$) and high specific energy (265.85 Wh kg$^{-1}$, based on the total mass of the active materials) in both half and full cells. This strategy of utilizing nanotwinned-alloy collectors has also been validated in the case of Cu-Ag alloys, presenting significant opportunities for the design of alloy collectors.

## Results

### Characteristics of the AlSi alloy collector

Figure 1a shows a typical process for preparing an AlSi alloy collector. Specifically, a conventional metallurgical process is employed to produce an as-cast AlSi alloy with industrial dimensions (Supplementary Fig. 1). Basically, the slow cooling during casting and the low solubility of Si (below 1.6 at.%) in Al under normal pressure conditions lead to the formation of large eutectic Si precipitates in grain boundaries (Supplementary Fig. 2). In contrast, the solubility of Si in the Al matrix increases with increasing pressure, and it is theoretically expected to reach 10 at.% (10.4 wt.%) at 5 GPa (Supplementary Fig. 3 and Supplementary Note 1). Thus, eutectic Si can be fully dissolved in the Al matrix, forming a metastable supersaturated solid solution[25]. Subsequently, the high-pressure-based AlSi samples (HP-AlSi) are subjected to low-temperature aging and JH treatment, enabling the precipitation of Si nanoparticles from the Al matrix to obtain typical high-pressure artificial aging based AlSi (HPA-AlSi) and HPJH-AlSi samples. The grain sizes of the above samples with five different alloy treatment processes have been shown in Supplementary Fig. 4. Furthermore, according to X-ray diffraction (XRD) analysis, the Si peaks of HPA-AlSi and HPJH-AlSi are significantly broader than the sharp Si peaks in the as-cast AlSi alloy, suggesting the formation of nanoscale Si from the metastable supersaturated solid solution (Supplementary Fig. 5).

To verify the structural variations in different samples induced by the HPJH, micromorphological characterization is conducted on both the HPA-AlSi and HPJH-AlSi alloys. According to transmission electron microscopy (TEM) analysis, the Si particles in both samples exhibit nearly identical sizes, with average particle sizes of approximately 42 and 41 nm for the HPJH-AlSi and HPA-AlSi samples, respectively (Fig. 1b, Supplementary Figs. 6 and 7a). However, the drastic change in the free energy of the supersaturated solid solution alloy during JH treatment results in the transient precipitation of solute Si, which leads to a significant increase in the proportion of NT-Si in the HPJH-AlSi sample, reaching a high value of 82.7%[22] (Fig. 1c). The crystal structure models of the NT-Si phase and Al matrix in the HPJH-AlSi alloy are shown in Fig. 1d. These structures are confirmed to exhibit typical twin strip textures by magnifying TEM images (Fig. 1e). As indicated by the corresponding energy-dispersive X-ray spectroscopy (EDS) mapping, the clear boundary between the Al matrix and NT-Si reveals no elemental segregation (Fig. 1e). Furthermore, the high-angle annular dark field (HAADF)–scanning transmission electron microscopy (STEM) image of the Al matrix reveals an interplanar spacing of 0.234 nm, which is consistent with the (111) plane of Al (Fig. 1f). Additionally, the lattice fringes of the NT-Si clearly exhibit a mirrored orientation relationship along the (111) crystal plane under the [011] zone axis. These stripes are found for the (111) nanotwins in the HPJH-AlSi alloy (Fig. 1g), which aligns perfectly with the crystal structure shown in Fig. 1c. Conversely, no twin defects are observed in the Si nanoparticles for the HPA-AlSi alloy and the interfaces between Al matrix and NT-Si or Si nanoparticle are shown in Supplementary Fig. 7. The geometric phase analysis (GPA) images show that significant strain effects are introduced on NT-Si particles in the HPJH-AlSi alloy, with many high surface energy atoms, increasing the probability of Si serving as active sites[26] (Fig. 1h). However, based on electron backscatter diffraction (EBSD) analyses, the HPJH-AlSi alloy primarily comprises equiaxed grains with uniform size distribution, and there is no preferential orientation among neighboring grains (Supplementary Fig. 8 and Supplementary Note 2). This implies that no significant deformation occurs in the matrix grains.

### Na nucleation and growth

To investigate the effects of different collectors on Na nucleation and growth, 0.1 mAh cm$^{-2}$ Na is deposited on HPJH-AlSi, HPA-AlSi and Al samples. Bascially, the deposits obtained at low capacities for short periods are primarily determined by the nucleation process[15].

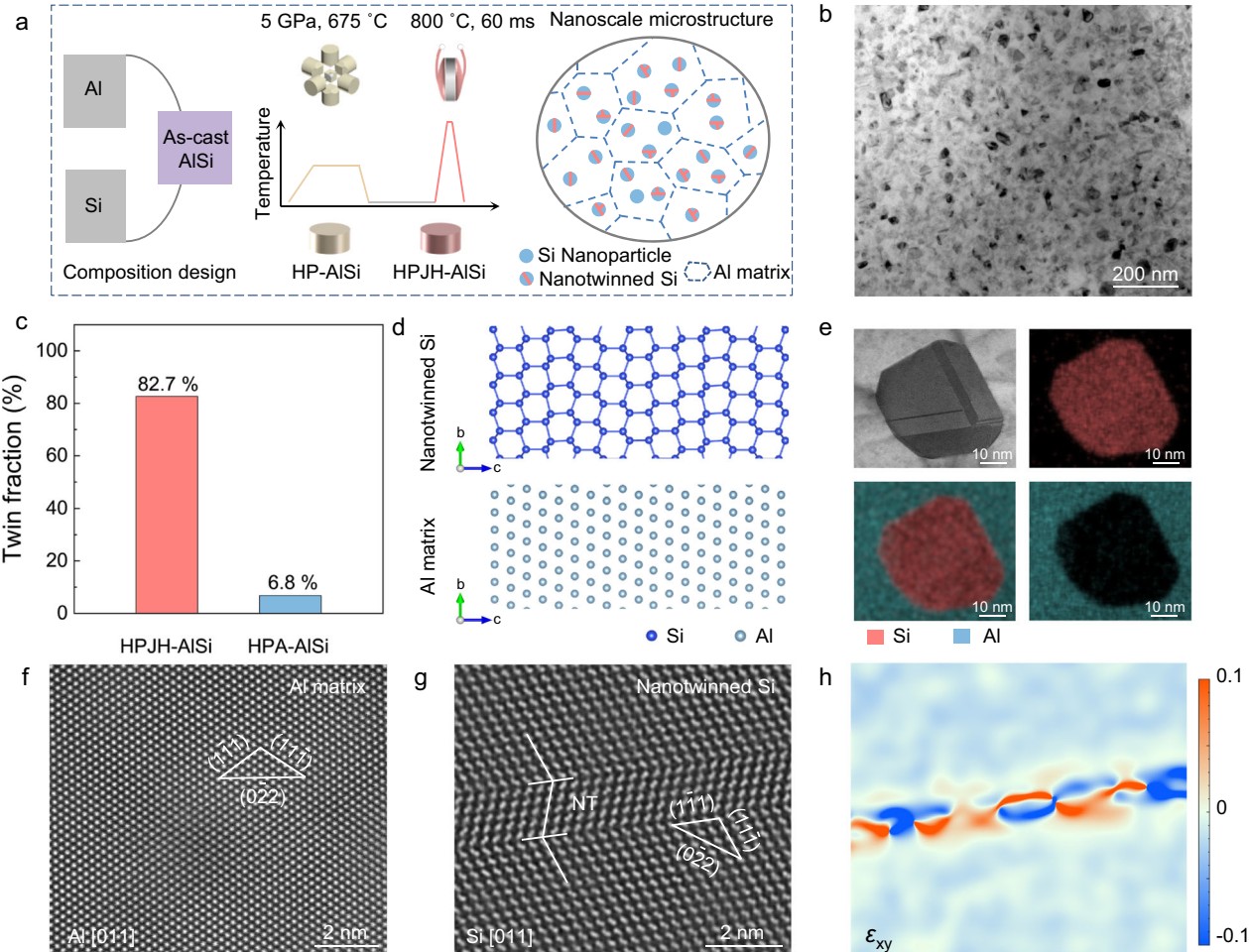

**Fig. 1 | Structural characteristics of Al-Si alloy. a** Schematic illustration of the preparation process of different Al-Si alloys. **b** TEM image of the HPJH-AlSi. **c** Fraction of NT-Si supported on HPJH-AlSi and HPA-AlSi statistically counted from TEM images. **d** Crystal structure of the NT-Si and Al matrix. The dark blue and light blue spheres denote Si and Al elements, respectively. **e** High-resolution transmission electron microscopy (HRTEM) images of NT-Si and the corresponding elemental mapping. **f** Atomic resolution HAADF image of the (111) Al taken along the [011] axis. **g** Atomic resolution HAADF image of the (111) NT taken along the [011] axis. **h** Geometric phase analysis map (horizontal shear strain, $\varepsilon_{xy}$ for HRTEM of NT-Si. Source data for (**c**) are provided as a Source Data file.

Scanning electron microscopy (SEM) images of Na deposited on the HPJH-AlSi alloy collector in the four electrolytes reveal uniformly distributed micron-sized spherical crystals (Fig. 2a–d). The spherical crystals that grew close to thermodynamic equilibrium have relatively small surface areas, hence reducing the occurrence of side reactions between Na and the electrolyte[27]. In contrast, the Na that is first nucleated on the HPA-AlSi and Al in the four electrolytes exhibits irregular and anisotropic dendritic morphologies (Fig. 2e–l and Supplementary Fig. 9). This outcome is mainly associated with the non-uniform and slow dynamics characteristics of the collectors[28,29].

To obtain a capacity equivalent to that of practical batteries, we use 1 M NaPF$_6$ in DIGLYME = 100 vol% ether-based electrolyte to investigate the influences of different morphologies of Na seeds on the growth characteristics of Na metal layers on the three collectors. The spherical Na crystal species on the HPJH-AlSi gradually agglomerates with increasing capacity, forming a uniform and dense planar morphology without a dendritic structure, as evidenced by the SEM images (Supplementary Fig. 10a). However, Na species are deposited on the HPA-AlSi and Al collectors as inhomogeneous bulk and randomly oriented filaments, respectively (Supplementary Fig. 10b, c). This finding demonstrates that the changes in different forms of Na during the deposition process is closely related to the substrate. Notably, the compositions of the solid−electrolyte interface (SEI) films on the different collector surfaces remain fundamentally unchanged after Na

deposition, as determined by X-ray photoelectron spectroscopy (XPS) analysis (Supplementary Fig. 11 and Supplementary Note 3). Therefore, it is believed that the collector composition, not the SEI film composition, governs the variation in the deposited morphology. In addition, after reaching the same deposition capacity, the Na plated on the HPJH-AlSi collector is thinner than its counterpart according to a side view of the morphology, suggesting that the Na layer plated on the HPJH-AlSi has a relatively high density and relatively low porosity, which is consistent with the phenomenon observed in the top view (Supplementary Fig. 10d–f). A similar trend is confirmed under various current densities (Supplementary Fig. 12). The summary plots clearly show that the utilization of HPJH-AlSi diminishes the initial porosity of Na deposition regardless of the current or capacity (Supplementary Fig. 13 and Supplementary Note 4). This reduction in porosity is attributed to the formation of spherical Na seeds during the initial Na nucleation process, facilitating an increasingly planar and dense accumulation of Na during subsequent deposition processes.

## In situ Na deposition
To gain profound insights into the Na deposition process and its related behavior, in situ TEM and molecular dynamics (MD) simulations have been employed to elucidate the dynamics of Na deposition, encompassing details such as the morphologies after nucleation and growth. In pursuit of in situ observations during Na deposition, an

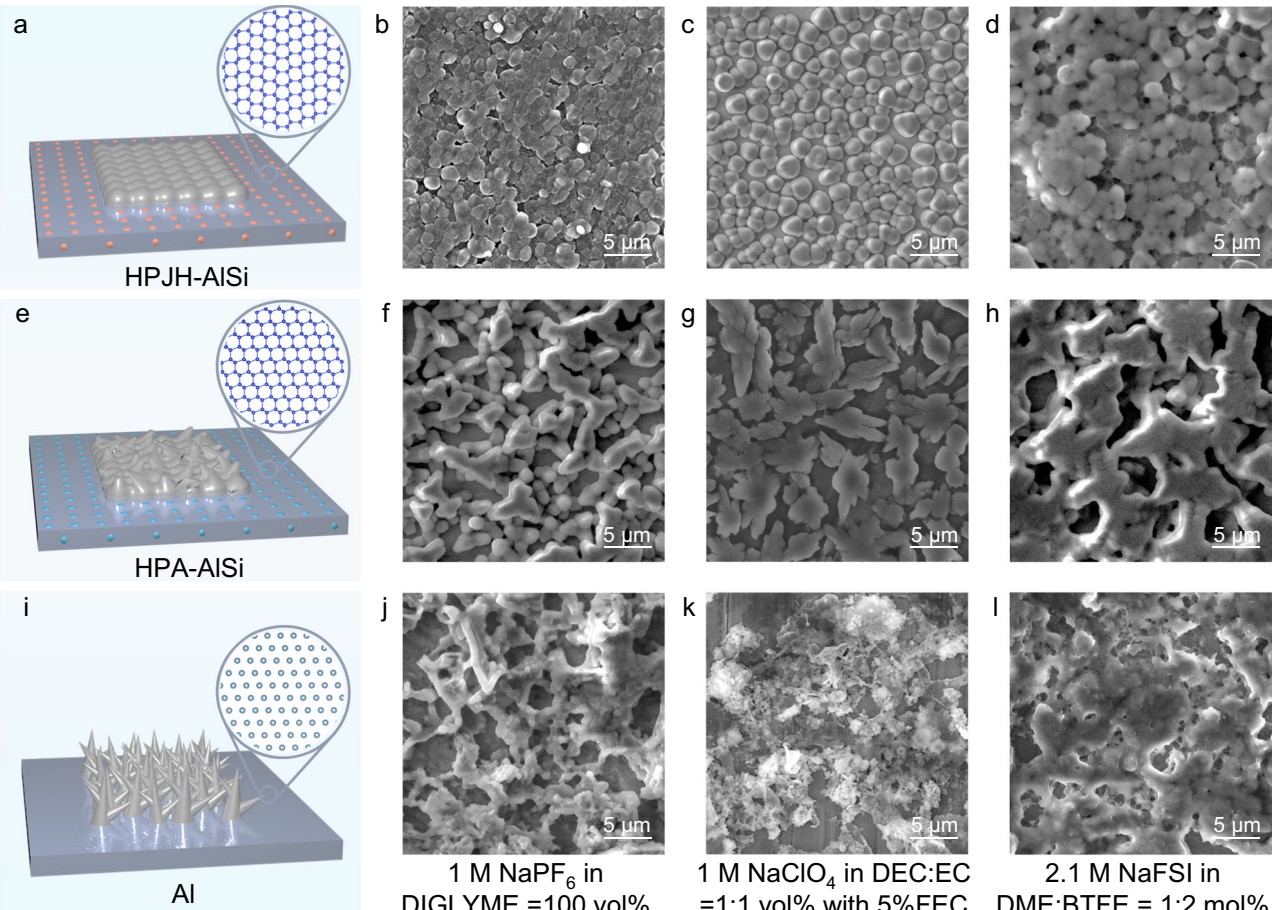

**Fig. 2 | Na nucleation and growth on different substrates. a, e, i** Schematic illustration of Na plating onto different collectors. The silver, gray, blue and pink denote deposited Na, Al matrix, nano-Si and nanotwinned-Si, respectively. In the atomic model diagram, dark blue spheres and light blue spheres represent Si and Al atoms, respectively. **b–d** Spherical Na deposited on HPJH-AlSi collector in different electrolytes. **f–h** Uneven Na deposited on HPA-AlSi collector in different electrolytes. **j–l** Dendritic Na deposited on Al collector in different electrolytes. The Na deposited morphologies on different collectors was measured after plating at a current density of 1 mA cm$^{-2}$ and a deposition capacity of 0.1 mAh cm$^{-2}$.

electrochemical deposition platform is meticulously constructed within an environmental transmission electron microscope by utilizing an all-solid-state cell assembly (Fig. 3a)[30]. Subsequently, Na deposition on HPJH-AlSi is executed by applying a negative potential (Fig. 3b–g and Supplementary Movies 1). Small Na spheres are observed at the intersection of HPJH-AlSi and Na in the initial stage, and the sizes of the spheres increases over time. Notably, the diameters of the Na spheres reach 400 nm after 150 s. Comparatively, typical Na dendritic morphologies on the HPA-AlSi and Al substrates, with cumulative lengths of 528 and 636 nm, respectively, are detected after the same deposition time, which is consistent with the SEM results (Fig. 3h, Supplementary Fig. 14 and Supplementary Movies 2, 3).

To further validate the superiority of the HPJH-AlSi surface, we conduct MD simulations at the nanoscale by analysing the morphological changes in Na on the HPJH-AlSi, HPA-AlSi, and Al surfaces, respectively (Fig. 3i–n and Supplementary Figs. 15, 16). For the HPJH-AlSi surface, the Na atoms are randomly and uniformly distributed, but they adhere closely to the surface due to the highly negative adsorption energy. The Na atoms begin to exhibit spherical morphologies at 600 ps. The spherical characteristics of the Na atoms are more evident after 800 ps than before. Finally, the spherical structures of the Na atoms merge into large spheres after 1000 ps. Comparatively, on the Al (111) surface, significant aggregation of Na atoms can be observed in the range of 0–400 ps. Subsequently, the Na atoms begin to exhibit dendritic features, and a distinct dendritic structure is

detected after 1000 ps. A similar process is observed for the HPA-AlSi surface. Namely, Na atoms begin to deviate from their spherical morphologies at the beginning, and then, they are prone to form columnar structures. At 1000 ps, the Na atoms form dense columnar structures. These results demonstrate significant differences in the deposition morphologies of Na atoms on various substrates, further confirming the unique advantages of the HPJH-AlSi surface.

## Mechanisms of NT-Si-induced Na deposition

According to classical nucleation theory, the initiation of a new phase necessitates surmounting an energy barrier, which can be discerned through electrochemical means[31]. Specifically, the results of electrochemical Na deposition, which is performed at different currents and voltage profiles, are plotted to investigate the Na dynamics. Notably, a conspicuous voltage dip is evident for the Al, HPA-AlSi, and HPJH-AlSi collectors, signifying the absence of a Na alloying reaction with Al and Si[32] (Supplementary Fig. 17a–c). Moreover, HPJH-AlSi exhibits slightly lower nucleation overpotential ($\eta_n$) and lower plating overpotential ($\eta_p$) values than those of Al and HPA-AlSi (Supplementary Fig. 17d, e), implying that the precipitated NT-Si exerts a discernible effect on Na dynamics[33]. Cyclic voltammetry (CV) tests further demonstrate that HPJH-AlSi has the highest peak current, suggesting that the presence of NT-Si contributed to the fastest Na dynamics in the alloy[34,35] (Supplementary Fig. 18). Additionally, throughout the Na deposition/stripping processes on all three collectors, no alloying peaks between Al or Si

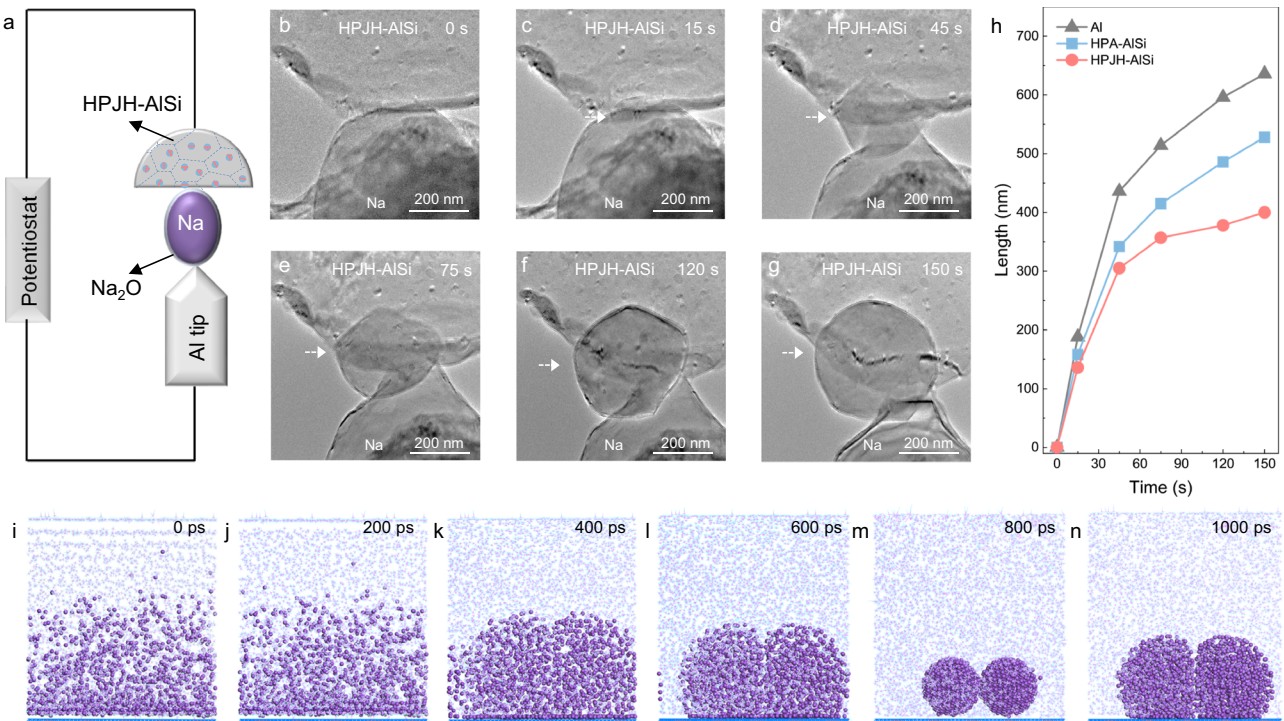

**Fig. 3 | In situ analysis of Na deposition. a** Schematic of the electrochemical deposition setup. **b**–**g** Time lapse TEM images of Na metal deposition on HPJH-AlSi collector. The nano cells were plated at a current density of 1 mA cm⁻² and measured under 25 °C. **h** Time-resolved evolution of Na size as a function of time during Na deposition on different substrates. **i**–**n** The temporal evolution of Na deposition on HPJH-AlSi obtained with MD simulations. The purple spheres, blue frames and pink frames represent deposited Na, Al matrix and nanotwinned Si, respectively. The MD data are provided at https://doi.org/10.24435/materialscloud:r1-e1. Source data for (**h**) are provided as a Source Data file.

and Na are observed. This confirms the inert nature of Al and Si in these electrochemical reactions, consistent with the voltage profiles during deposition. Furthermore, to reduce the effect of the SEI film on Na electrodeposition, linear sweep voltammetry (LSV) under fast current density regimes is implemented to evaluate Na transport from the bulk to the depositing current collector surface[36] (Supplementary Fig. 19a–c). The Na diffusion dynamics can be theoretically obtained according to the Randles–Sevcik equation[19]:

$$J_p = \left(2.65 \times 10^5\right) n^{\frac{3}{2}} S C_{Na}^* D_{Na}^* v^{\frac{1}{2}} \tag{1}$$

where $J_p$ is the peak current, $n$ is the electron number, $S$ is the electrode area, $D_{Na}$ is the Na diffusion coefficient, $C_{Na}$ is the Na-ion concentration in the electrochemical reaction, and $v$ is the scan rate. As shown in Eq. (1), $D_{Na}$ is positively correlated with the slopes of the $J_p/v^{1/2}$ curves (Supplementary Fig. 19d). By examining the variations in peak current $J_p$ values at scan rates of 5, 10, and 20 V s⁻¹, the slopes of the $J_p/v^{1/2}$ curves are plotted, as shown in Fig. 4a. The findings indicate that the $J_p/v^{1/2}$ slope of HPJH-AlSi exceeds that of HPA-AlSi and Al, indicating a significant increase in the Na diffusion rate in HPJH-AlSi compared to that in HPA-AlSi and Al. In addition, the mean square displacement (MSD) results from MD simulations show a sharp increase in the MSD of Na on the HPJH-AlSi alloy over time, indicating that Na undergoes rapid diffusion[37]. Specifically, the diffusion coefficients of Na on HPJH-AlSi, HPA-AlSi, and Al are $1.428 \times 10^{-3}$, $8.465 \times 10^{-4}$, and $6.196 \times 10^{-4}$ Å²/ps, respectively (Fig. 4b). It is confirmed that it can quickly migrate when Na is deposited on the HPJH-AlSi alloy surface, which is advantageous for the formation of spherical Na and for the dense growth of Na metal on the alloy collector[38].

To elucidate the role of alloying components in directing the crystal growth of deposited Na metals, the specific interaction and diffusion barriers for Na on decisive low-index crystal planes (including

Al(111), Si(111) and NT-Si(111)) are calculated for three different alloys: Al, HPA-AlSi, and HPJH-AlSi. Specifically, binding energy calculations show that Na is strongly coordinated with NT-Si, not with Al and nano Si (Supplementary Fig. 20). Concurrently, the charge density difference results show that the electron distribution is confined within the adsorption sites on the Al and nano Si surfaces, leading to a weakening of the coupling interactions; however, the charge transfer from Na to the NT-Si surface is relatively pronounced (Supplementary Fig. 21). Therefore, the strong and directed Na coordination induced by NT-Si facilitates the transport of additional Na from the electrolyte environment to the alloy collector surface[39,40]. Additionally, the diffusion barriers of Na between neighboring adsorption sites on the Al(111), Si(111) and NT-Si(111) facets are determined to be 0.86 eV, 0.62 eV, and 0.48 eV, respectively (Fig. 4c and Supplementary Fig. 22). This finding implies that the Na atoms on the NT-Si(111) surface exhibit extensive migration[41]. Importantly, the morphological changes in Na are governed by the competing mechanisms of charge transfer and diffusion limitations during electrochemical deposition[42]. Dendritic growth tends to occur under diffusion control (high reaction rate and low diffusion rate), while reaction control (low reaction rate and high diffusion rate) dominates the formation of spherical deposition[42]. To examine the balance between the surface diffusion rate and the electrochemical reaction rate, a nondimensional electrochemical Damköhler number (Fig. 4d, Supplementary Note 5 and Supplementary Table 1), $D_a$, can be defined as the ratio of the electrochemical reaction rate, $k_e$, to the surface diffusion rate, $k_d$:[41]

$$D_a = \frac{k_e}{k_d} \tag{2}$$

The $D_a$ for Al is much larger than 1, implying that the electrochemical reaction occurs at a much faster rate than surface self-

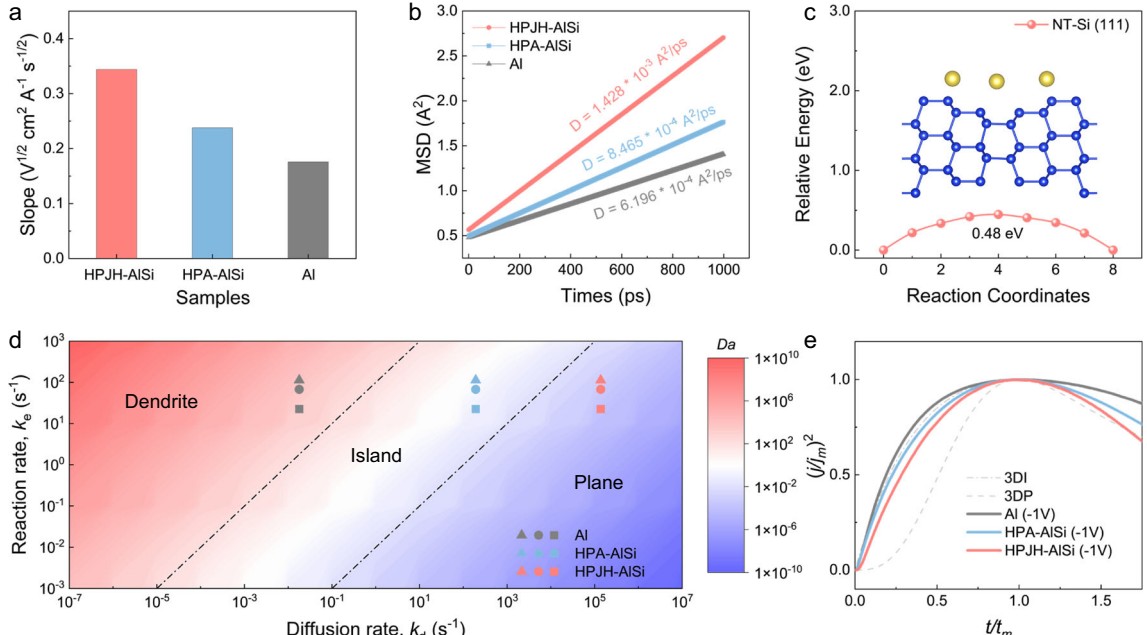

**Fig. 4 | Sodiophilic mechanism. a** The slope value of the linear relationship between LSV peak current and scan rate. **b** MSD simulation of Na on different collectors. **c** The diffusion pathway and calculated energy profile of Na along the diffusion path of Na on the NT-Si. The blue spheres and yellow spheres represent Si and Na atoms, respectively. **d** Deposition morphology phase map as a function of diffusion. Thus, dendritic structures are expected. The $D_a$ value of HPA-AlSi is approximately equal to 1, which suggest that the rate of surface self-diffusion is very close to the rate of electrochemical reactions, and columnar-island-like structures are inevitable. In contrast, the $D_a$ of the HPJH-AlSi is much less than 1, which indicates that the electrochemical reaction rate is slower than the surface self-diffusion rate, making spherical planar deposition dominant.

We utilized chronoamperometry (CA) to further investigate the nucleation and growth of Na (Supplementary Fig. 23 and Supplementary Note 6). In brief, all transients display a peak current ($j_m$), signifying characteristic 3D nucleation and growth[15]. Moreover, the current-time transients are first normalized using $j_m$ and the corresponding $t_m$ and then compared with the Scharifker–Hills model[43]. According to this model, nucleation occurs in two modes: three-dimensional instantaneous (3DI) or three-dimensional progressive (3DP). The classification depends on whether new nuclei emerge abruptly at onset or gradually over time. The mathematical expressions are as follows:

$$3DI : \left(\frac{j}{j_m}\right)^2 = 1.9542\left(\frac{t}{t_m}\right)^{-1}\left\{1 - exp\left[-1.2564\left(\frac{t}{t_m}\right)\right]\right\}^2 \quad (3)$$

$$3DP : \left(\frac{j}{j_m}\right)^2 = 1.2254\left(\frac{t}{t_m}\right)^{-1}\left\{1 - exp\left[-2.3367\left(\frac{t}{t_m}\right)^2\right]\right\}^2 \quad (4)$$

The normalized dimensionless transients are depicted in Fig. 4e, indicating that the nucleation on the Al collector closely aligns with the expected response for 3DI. Obviously, the nucleation sites are extremely limited and become depleted in the early stages of the process. In contrast, the observed nucleation behavior on the HPJH-AlSi collector somewhat adheres to a hybrid model—between 3DI and 3DP for $t < t_m$ and 3DP for $t > t_m$. In this scenario, nucleation sites are progressively activated, and the process is accompanied by nuclei growth. Additionally, the fitting results for the HPA-AlSi collector are in between

those for the Al collector and the HPJH-AlSi collector, indicating that the nucleation sites are relatively limited, the Na dynamics are relatively sluggish. Therefore, the amount of diffusion-controlled growth during this process is higher than that of the HPJH-AlSi collector.

Based on the aforementioned results, we schematically summarize the potential nucleation and growth mechanisms involved. In the case of a sodiophobic Al collector following the 3DI model, Na nuclei emerge randomly on limited nucleation sites within a short period. The nuclei tend to grow in their initial positions without the formation of new nuclei due to the elevated nucleation barrier and the slow Na diffusion dynamics. Subsequent growth is propelled by the 3D volume expansion of this structure. For the HPJH-AlSi alloy following the 3DI + 3DP model, NT-Si with high sodiophilicity can offer additional nucleation sites that experience progressive activation. Furthermore, these particles are uniformly distributed and characterized by low diffusion barriers, which guide the subsequent growth of Na spheres, resulting in the formation of a relatively flat Na deposition layer.

## Electrochemical performance

To evaluate the effect of the HPJH-AlSi alloy collector on reversible Na migration, plating/stripping measurements have been performed in symmetric and asymmetric cells. The Na||HPJH-AlSi cells, employing a current density of 3 mA cm$^{-2}$ and a capacity of 1 mAh cm$^{-2}$, demonstrate a stable plating/stripping process over 4000 cycles (cumulative capacity of 4 Ah cm$^{-2}$) with an average Coulombic efficiency of 99.71% (Fig. 5a). Moreover, the potential curves corresponding to these cycles are stable, as confirmed by a minimal voltage polarization (~43.3 mV), suggesting the robust stability of the precipitated NT-Si anchoring sites (Fig. 5b). Comparatively, cells containing Al, HP-AlSi, As-cast AlSi, and HPA-AlSi exhibit subpar Na plating/stripping performance and great voltage hysteresis, resulting in rapid deterioration (Supplementary Figs. 24, 25). The pronounced fluctuations in the signals observed before cell failure originate from the random reconnection of inactive Na, which is closely associated with nonuniform deposition[35].

Damköhler number. **e** Plots of $t/t_m$ and $(j/j_m)^2$ in comparison with theoretical 3D models for a Na cell using Al, HPA-AlSi and HPJH-AlSi collector. The DFT data are provided at https://doi.org/10.24435/materialscloud:r1-e1. Source data are provided as a Source Data file.

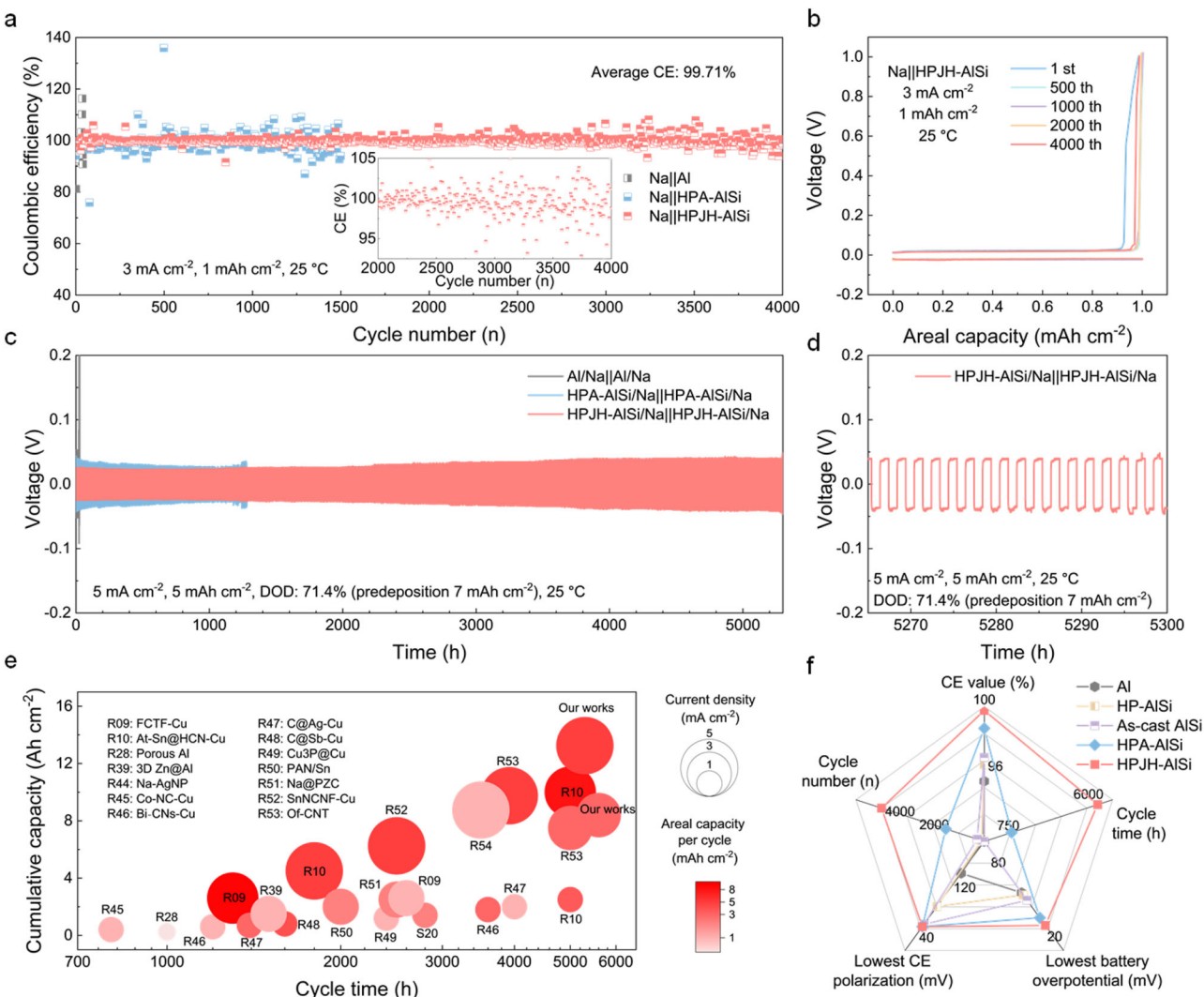

**Fig. 5 | Asymmetric and symmetric cell performances. a** Coulombic efficiencies of Na plating/stripping on different collectors at current densities of 3 mA cm⁻² and areal capacities of 1 mAh cm⁻² and the magnified view of selected cycle numbers between 2000th and 4000th cycle for the Na || HPJH-AlSi (Inset). **b** Voltage-capacity profiles of HPJH-AlSi at 3 mA cm⁻² and 1 mAh cm⁻². **c** Voltage profiles of symmetric cells assembled with different collectors at 5 mA cm⁻² and 5 mAh cm⁻². **d** Magnified view of selected cycle times between 5265 and 5300 h for the

symmetric cell assembled with HPJH-AlSi. All the tests were measured at 25 °C using the 1 M NaPF₆ in diglyme =100 vol% electrolyte. **e** A comparative analysis of cumulative capacity and plating/stripping time in the recent demonstration of Na symmetric cells with our work. Detailed information for each demonstration are provided in Supplementary Table 2. **f** Radar plots of the five main electrochemical properties of different collectors. Source data are provided as a Source Data file.

Interestingly, the HPJH-AlSi/Na||HPJH-AlSi/Na symmetric cell achieves a rare and long cycle approaching 5600 h under a high current density of 3 mA cm⁻² and a capacity of 3 mAh cm⁻². The cumulative capacity of this cell even reaches an extremely high value of 8.4 Ah cm⁻² (Supplementary Fig. 26). Moreover, at a higher current density of 5 mA cm⁻² and a capacity of 5 mAh cm⁻², the HPJH-AlSi/Na || HPJH-AlSi/Na symmetric cell can maintain stable Na plating/stripping for up to 5300 h, with a cumulative capacity of 13.25 Ah cm⁻² (Fig. 5c), which is higher than most of reported values (Fig. 5e, Supplementary Note 7, and Supplementary Table 2)[9,10,28,39,44–53]. Furthermore, even with the HPJH-AlSi/Na || HPJH-AlSi/Na symmetric cell undergoing a long plating/stripping duration of 5300 h under severe conditions, the overpotential remains consistently low (Fig. 5d). This performance overwhelms those of batteries containing Al, HP-AlSi, as-cast AlSi, and HPA-AlSi (Supplementary Fig. 27). Even under more stringent conditions—8 mA cm⁻² current density, 8 mAh cm⁻² capacity, and 80% depth of discharge—the cell maintains stable Na plating/stripping for up to 800 h (Supplementary Fig. 28). Moreover, the HPJH-AlSi/Na||HPJH-AlSi/Na symmetric cell reveals good rate capability under increasing

current densities approaching 10 mA cm⁻² (Supplementary Fig. 29), whereas the voltage–time curve of the Al/Na ||Al/Na cell suffers from severe fluctuations. The radar chart summarizing the performance of the five current collectors illustrates the capabilities of the HPJH-AlSi collector are better than those of the others (Fig. 5f). Undoubtedly, the performance metrics of the HPJH-AlSi collector are superior to those of other collectors in terms of five critical parameters (Supplementary Table 3). Additionally, after 10 cycles at a current density of 5 mA cm⁻² and a capacity of 5 mAh cm⁻², the HPJH-AlSi current collector still retains a dense distribution of nanotwinned Si and exhibits a well-preserved Na plating/stripping morphology (Supplementary Figs. 30, 31). Comparatively, the pronounced dendritic and nonuniform distribution of Na on both the Al and HPA-AlSi collectors exacerbates parasitic reactions (Supplementary Fig. 31). XPS analysis further confirms that the HPJH-AlSi/Na electrode exhibits the lowest oxide formation, which correlates with its minimal parasitic reactions and superior performance (Supplementary Fig. 32). The Al/Na electrode shows the highest oxide content, consistent with its poor performance due to severe surface reactions. In comparison, the HPA-AlSi/Na

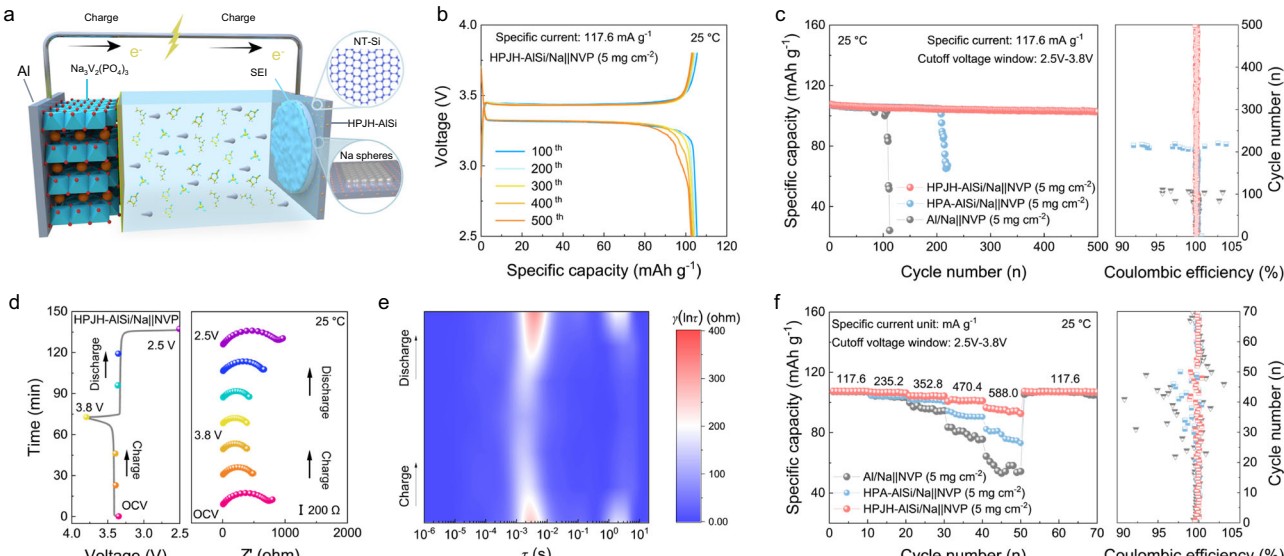

**Fig. 6 | Electrochemical performance of full cells. a** Schematic illustration of the internal structure of the full cells. **b** Typical voltage profiles of HPJH-AlSi/Na ‖ NVP cells at the specific current of 117.6 mA g⁻¹. **c** Long-term cyclic stability of the different full cells and the corresponding Coulombic efficiency (Right panel). **d** In situ EIS results of the HPA-AlSi/Na ‖ NVP. **e** Corresponding contour plots of the calculated DRT results from HPJH-AlSi/Na ‖ NVP. **f** Rate capability of the different full cells and the corresponding Coulombic efficiency (Right panel). All full cells were tested under the cut-off voltage window of 2.5–3.8 V and NVP loadings of 5 mg cm⁻². All the tests were measured at 25 °C. Source data for (**b**–**f**) are provided as a Source Data file.

electrode exhibits a moderate level of oxide formation, corresponding to its performance between HPJH-AlSi/Na and Al/Na. Furthermore, to demonstrate the superiority of the HPJH alloying strategy, we prepare pure Al coated with a Si layer and AlSi alloy treated with normal pressure solid solution combined with JH treatment for comparison. The weak bonding between the Si coating and the Al substrate causes detachment during deposition and stripping, resulting in the Na plating/stripping performance of the Si-coated-Al symmetric cells being inferior (Supplementary Figs. 33–35 and Supplementary Note 8). In addition, AlSi alloys with normal pressure solid solution combined with JH treatment have a large amount of Si remaining undissolved in the Al substrate due to the limited solubility increase at normal pressure (Supplementary Fig. 36a–c). During JH, only a small fraction of Si dissolves and subsequently precipitates as Si particles. Meanwhile, the precipitated Si particles are relatively small, about 20 nm, and no twins are observed inside the Si particles (Supplementary Fig. 36d–f). As a result, the current collector is only able to 75 h at a current density of 5 mA cm⁻² and a capacity of 5 mAh cm⁻², which is significantly lower than the performance of HPJH-AlSi. (Supplementary Fig. 36g, h).

## Full cell performance

To highlight the applicability of the HPJH-AlSi alloy, we investigate a full cell (negative/positive (N/P) ratios: 3.7) comprising an HPJH-AlSi alloy negative electrode (2 mAh cm⁻² as-deposited Na) and a commercial Na₃V₂(PO₄)₃ (NVP) positive electrode (5 mg cm⁻²) (Fig. 6a and Supplementary Fig. 37). In the form of a coin cell, HPJH-AlSi/Na ‖ NVP exhibits a lower polarization voltage and higher specific capacity than the Al/Na ‖ NVP and HPA-AlSi/Na ‖ NVP during cycling, indicating enhanced reversibility[9] (Fig. 6b and Supplementary Fig. 38a–c). In particular, the capacity retention ratio of 95.4% remains greater than 500 cycles at 117.6 mA g⁻¹ for the HPJH-AlSi/Na ‖ NVP, whereas Al/Na ‖ NVP and HPA-AlSi/Na ‖NVP fail at the 112th and 220th cycles, respectively (Fig. 6c, Supplementary Fig. 38a–c and Supplementary Note 9). These outstanding results obtained using the HPJH-AlSi alloy outperform previously published findings (Supplementary Table 4). Importantly, compared with previous reports on Na full cells, our HPJH-AlSi/Na ‖NVP demonstrates a very competitive specific energy of 265.85 Wh kg⁻¹ at a specific power of 290.55 W kg⁻¹ (based on the total

mass of the active materials), as shown in Supplementary Fig. 38d, Supplementary Note 10 and Supplementary Table 5.

To evaluate the interfacial changes in various full cells, we perform in situ electrochemical impedance spectroscopy (EIS) measurements (Fig. 6d and Supplementary Fig. 39). The impedance of the HPJH-AlSi/Na ‖ NVP full cell exhibits a notable decrease from the initial state to the beginning of the discharge process. Moreover, the final resistance after discharge closely resembles that of the initial state, suggesting good reversibility in the HPJH-AlSi/Na ‖ NVP full cell. In turn, the resistance values of both the Al/Na ‖ NVP and HPA-AlSi/Na ‖ NVP full cells significantly increase within a single cycle, indicating that irreversible dendrite growth and SEI deterioration typically occur[54]. To distinguish the multiple reactions and processes represented by the impedance values within the cell, an analysis of the corresponding distributions of relaxation times (DRT) is conducted using DRTtool[55]. Specifically, τ1, τ2, and τ3 correspond to the ion transport process on the SEI, the charge transfer reaction, and the mass transfer in the electrolyte and electrode phases, respectively[56] (Fig. 6e, Supplementary Figs. 40–41 and Supplementary Note 11). For Al/Na ‖ NVP and HPA-AlSi/Na ‖ NVP, the peaks of τ1, τ2, and τ3 are notably high, signifying significant changes during cycling. This finding suggests that the ion and electron transport undergo substantial changes due to dendrite formation and SEI deterioration[54]. However, the intensities of the τ1, τ2, and τ3 peaks are small in HPJH-AlSi/Na ‖NVP, with slight variations during cycling. This finding reflects the homogeneous deposition/stripping process of Na, leading to the stabilization of the SEI and the idealization of charge transfer[56].

In terms of practical implications, full cells are frequently serviced in challenging environments characterized by various rates. Notably, the discharge specific capacities of the HPJH-AlSi/Na ‖NVP full cells are remarkable, with specific capacities of 107.1, 106.3, 104.3, 101.1, and 92.7 mAh g⁻¹ as the specific current increases from 117.6 to 588.0 mA g⁻¹ (Fig. 6f). These values surpass those of comparable full cells, underscoring the good rate capabilities of HPJH-AlSi/Na ‖NVP full cells. This improvement can be attributed to the rapid diffusion dynamics of Na from NT-Si during plating/stripping, which promotes the densification and homogeneity of the Na morphology (Supplementary Fig. 42). These notable results support the potential

applicability of the HPJH-AlSi alloy in energy-dense and practically viable energy storage devices.

## Discussion

To clearly understand the intrinsic properties of the current collector before and after alloying, we conducted tests on the electrical conductivity and mechanical properties of both pure Al and HPJH-AlSi alloy (Supplementary Fig. 43, Supplementary Note 12 and Supplementary Table 6). Compared with pure Al, the electrical conductivity of the HPJH-AlSi alloy can remain the same order of magnitude (Supplementary Fig. 43a), indicating that its application in Na metal batteries is not significantly compromised. More importantly, the HPJH-AlSi alloy shows substantial improvements in mechanical properties in contrast to pure Al (Supplementary Fig. 43b). These enhancements are crucial for maintaining the structural integrity of the current collector during Na plating/stripping, particularly under the repeated processes of Na deposition and stripping. Furthermore, to extend the potential applications of this method on coherent interface induced Na deposition, we also synthesize the HPJH-CuAg alloy with a nanotwined Ag structure (Supplementary Fig. 44). As a result, the cycle lives of Na symmetric cells employing HPJH-CuAg reach 250 h, but the cells do not exhibit dendritic morphologies (Supplementary Fig. 45). Similarly, the low binding energies of Na to NT-Ag(111) indicate that NT-Ag is a nucleation site in the HPJH-CuAg collectors, which is the reason for the good performance of the alloy collector (Supplementary Fig. 46a). Additionally, the calculated values of the low diffusion barrier between Na and NT-Ag(111) indicate that NT-Ag can effectively enhance Na dynamics and promote the dendrite-free deposition of Na (Supplementary Fig. 46b).

In summary, differing from traditional methods, we have designed a nanotwinned nucleation site alloy collector using the HPJH method to enhance the Na deposition dynamics. This approach transforms Na deposition from diffusion-controlled process to reaction-controlled process, facilitating spherical Na deposition and dendrite-free growth. As a result, the cells equipped with HPJH-AlSi have achieved high capacities and a long plating/stripping lifespans under demanding conditions, such as high utilization and large-rate circulation. This nanotwinned alloy strategy is equally scalable to other alloy materials, providing a significant impetus for the advancement of of dendrite-free metal batteries.

## Methods

Figure 1a was created using Cinema 4D (C4D, R20) and PowerPoint (PPT, 2021). Figures 2a, e, f, and 6a were created using Cinema 4D (C4D, R20) and Vienna ab initio simulation package (VASP, 6.4.0). Figure 3a was created using PowerPoint (PPT, 2021). The experimental data extraction and processing were performed using Origin 2022 software.

### Preparation of the alloy collector

As-cast Al-10Si (wt.%, thereafter in wt.%) alloys were fabricated through a conventional smelting method by utilizing 99.99 wt.% pure Al and 99.99 wt.% pure Si which were bought from Aladdin Reagent (Shanghai) Co., Ltd., China. Subsequently, the resultant samples were machined into cylinders with diameters of 56 mm and heights of 20 mm for HP solid-solution treatment, which was conducted in a cubic-anvil high-volume press equipped with six rams. The treatment parameters included a specific pressure, temperature, and duration of 5 GPa, 675 °C, and 30 min, respectively. The resulting material was denoted as HP-AlSi. Then, the HP-AlSi samples underwent an additional JH treatment (Ultrafast High-Temperature Furace 2023 A, Particle Precision instruments Co. Ltd, jilin Province, China) at 800 °C for 60 ms. Before the JH treatment, the samples were polished on both sides and positioned between two copper sheets to enhance JHed effects. Finally, the JH-treated samples were promptly quenched in cold water and named HPJH-AlSi. In addition, the samples subjected to normal artificial ageing at 180 °C for 30 min after HP treatment were referred to as HPA-AlSi for comparison.

Similarly, as-cast Cu-10Ag alloys were derived through the same smelting process, employing 99.99 wt.% pure Cu and 99.99 wt.% pure Ag bought from Aladdin Reagent (Shanghai) Co., Ltd., China. In the subsequent HP treatment, the heating temperature for the HP-CuAg alloy was set to 850 °C. The remaining high-pressure parameters aligned with those applied to the HP-AlSi alloy. Furthermore, the JHed preparation parameters for the HPJH-CuAg alloy were consistent with those used for the HPJH-AlSi alloy.

### Microstructural characterization

XRD analyses of the crystallographic phases and chemical compositions were conducted using a Rigaku D/MAX-2005/PC instrument. Cu Kα radiation ($\lambda = 1.5406$ Å) was used, with a step scan of 0.02° per step and a scan rate of 4° min$^{-1}$. The microstructural morphologies and elemental compositions were observed through SEM (FEI Helios G4CX) at an accelerating voltage of 5 kV for SEM imaging and at 20 kV for EDS mapping. A working distance of 8 mm was maintained during imaging. The dwell time for each SEM image was set to 5 s, while for EDS mapping, the dwell time was 2 min. These settings ensured accurate data collection and minimized the risk of Na melting during the analysis. TEM analysis was performed using an environmental transmission electron microscope (ETEM; Titan ETEM G2) operating at 300 kV. To eliminate the effects of oxidation, the surface layers of the samples with thicknesses of 0.5 mm were removed by mechanical polishing. The elemental compositions of the materials were determined through XPS, utilizing a Thermo Scientific Escalab 250Xi X-ray photoelectron spectrometer with a monochromatic Al Kα X-ray source. The grain structure, orientation, and phase distribution of the alloy were analyzed over a relatively large area using EBSD. For the EBSD sample preparation, electrolytic polishing was conducted at temperatures below 10 °C in a mixed solution of 10 vol% HClO$_4$ (purity 70.0–72.0%, Aladdin,) and 90 vol% C$_2$H$_6$O (purity 99.5%, Aladdin). The EBSD patterns were acquired at an acceleration voltage of 20 kV. The obtained results were post-processed using HKL Channel 5 software. Additionally, during the electrode testing process, the samples were transported using an Ar gas filled transfer box to ensure they remained uncontaminated and stable during the transfer.

### Cell assembly and electrochemical measurements

For electrochemical measurements of Na plating/stripping, The CR2032-type coin cells with 316 stainless steel sping (Diamerter 15.4 mm × Thickness 1.1 mm, MTI) were assembled in an Ar-filled glovebox (O$_2$ < 0.1 ppm, H$_2$O < 0.1 ppm). All electrochemical data were collected by a multichannel cell testing station (CT2001A, LAND) and measured at 25 °C. The Na plating and stripping processes were investigated using half cells, wherein various AlSi alloys served as the working electrode, and Na (purity 99.7%, Aladdin) foil was employed as the counter electrode. A glass fiber with 47 mm diamater, 670 µm thickness and 2.7 µm pore size (GFD Wattman, UK) was utilized as the separator, and approximately 100 µL of electrolyte was used. The following high-purity electrolytes (≥99%) sourced from DoDoChem were used for Na nucleation and growth: 1 M NaPF$_6$ in diglyme = 100 vol%, 2.1 M NaFSI in DME:BTFE = 1:2 mol%, 1 M NaClO$_4$ in DEC:EC = 1:1 vol% with 5% FEC, and 1 M NaClO$_4$ in PC = 100 vol% with 5% FEC. The assembled half cells underwent three precycling cycles between 0.01 and 1 V (versus Na$^+$/Na) at 1 mA cm$^{-2}$ to stabilize the SEI and eliminate surface contamination in the CE tests. Subsequently, specific quantities of Na were first deposited on different alloy collectors under galvanostatic conditions for a controlled time and then stripped to 1 V for each cycle.

For the symmetric cells, a specific amount of Na was initially deposited under 1 M NaPF$_6$ in diglyme = 100 vol% (purity 99%,

DoDoChem) at a current density of $1\,mA\,cm^{-2}$ onto various current collectors within half cells. Then, these cells were disassembled within a glove box to acquire Na-loaded electrodes. Symmetric cells were then constructed utilizing two identical predeposited Na electrodes for the plating/stripping measurements. The Na load before deposition was maintained at $5\,mAh\,cm^{-2}$ or $7\,mAh\,cm^{-2}$ for symmetric cells, with a plating/stripping capacity of $3\,mAh\,cm^{-2}$ or $5\,mAh\,cm^{-2}$, respectively.

For the full-cell positive electrode, a slurry containing 80 wt.% sodium vanadium phosphate (NVP, purity 99%, MTI) as the active material, 10 wt.% Super P (purity ≥99 %, MTI), and 10 wt.% polyvinylidene difluorides (PVDF, purity ≥ 99 %, $M_w = 1,000,000$, MTI) as the binder in N-methyl pyrrolidone (NMP, purity 99.9%, Aladdin) was prepared. Afterwards, the slurry was single-side coated onto an Al foil (purity 99%, 12 μm, MTI) by automatic coating machine (MRX-TMH250, Shenzhen Mingruixiang Automation Equipment Co., Ltd.) and dried at 80 °C for 12 h in a vacuum oven. The electrode was cut into diameter of 16 mm using manual cutting machine (MSK-T10, MTI). The mass loading of the active materials in the positive electrodes was controlled at $5\,mg\,cm^{-2}$. The specific capacities were computed based on the mass of NVP in the positive electrode. The full cell operated within a working potential window of 2.5 to 3.8 V. The electrolyte was1 M $NaClO_4$ in DEC:EC = 1:1 vol % with 5% FEC. An amount of $2\,mAh\,cm^{-2}$ of Na was deposited on various current collectors, serving as the negative electrode in the full cells. In situ EIS measurements of the full cells were conducted using a Biologic VMP3 electrochemical workstation with an amplitude of 10 mV. The applied signal was potentiostatic, The EIS test was set with 6 points per decade of frequency, and the data were recorded at 21 min intervals. The frequency range for the measurements spanned from 100 kHz to 0.1 Hz. Additionally, charge and discharge EIS tests were performed in galvanostatic mode with a specific current of $117.6\,mA\,g^{-1}$. Prior to the EIS measurement, the sample was held at a quasi-stationary potential to ensure stability, with the quasi-stationary potential being applied for 5 min. All EIS measurements were conducted at 25 °C.

## In situ electrochemical deposition

To observe the in situ Na deposition process of HPJH-AlSi, HPA-AlSi and Al, an electrochemical deposition platform was developed within an environmental transmission electron microscope. HPJH-AlSi, HPA-AlSi or Al was used as the positive electrode, and Na metal was scratched onto a sharp Al tip as the counter electrode. Notably, the naturally formed $Na_2O$ layer on the surface of the Na metal served as the electrolyte. The electrochemical device was integrated into a TEM with a scanning tunneling microscope (STM) holder (Pico Femto FE-F20) in a glove box. Subsequently, the assembly was sealed in a custom airtight bag filled with dry argon gas and transferred to the ETEM. The total exposure time to air was kept under 2 s, limiting the thickness of the oxide layer on the metal Na surface. After applying a negative bias (0 V to −5 V) and conducting at a current density of $1\,mA\,cm^{-2}$ to the HPJH-AlSi, HPA-AlSi or Al against the Na electrode, the electrochemical deposition process was initiated.

## Density functional theory (DFT) calculations

The binding energy calculations were conducted using the Vienna ab initio simulation package (VASP) while employing the projector augmented wave (PAW) methodology[57]. The generalized gradient approximation (GGA) with Perdew–Burke–Ernzerhof (PBE) exchange-related generalized functions was utilized[58]. The cut-off energy for kinetic energy was set to 400 eV, and the Monkhorst–Pack method with a $5 \times 5 \times 1$ k-mesh in the Brillouin zone was employed. The energy convergence tolerance was maintained at $1.0 \times 10^{-5}$ eV/atom, and the force certification was set to 0.03 eV/Å. The binding energy ($\Delta E_b$) was determined using the following formula: $\Delta E_b = E_{adsorbate+support} - (E_{support} + E_{adsorbate})$, where $E_{adsorbate+support}$ is the total energy of the materials with adsorbed molecules and $E_{support}$ and $E_{adsorbate}$ are the

energies of substrates and the chemical potential of Na, respectively. The transition state was located via the climbing image nudged elastic band (CINEB) method[59].

## MD simulations

The calculations were carried out using the Forcite software package. To model the potential energy surface (PES), the Universal Force Field (UFF) was employed[60]. Atomic charges were assigned through the Charge Equilibration (Qeq) method with a tolerance of $5 \times 10^{-4}$ e. The Ewald summation method was utilized to calculate non-bonded interactions, achieving a precision of $10^{-5}$ kcal/mol. During the optimization procedure, all atoms were allowed to relax, with convergence criteria set at $2 \times 10^{-5}$ kcal $mol^{-1}$ for the calculated energy and 0.001 kcal/mol/Å for forces. After the geometry optimization, a 100 ps constant volume, constant temperature (NVT) molecular dynamics (MD) simulation was performed, followed by a 150 ps constant pressure, constant temperature (NPT) simulation, both using a time step of 1 fs. Data collection occurred during the 1 ns production phase. The Andersen thermostat was applied to control the simulated temperature at 300 K[61], whilst the Berendsen barostat was applied for regulating the simulated pressure throughout the simulations[62].

## Data availability

The datasets generated during and/or analyzed during the current study are available from the corresponding author on request. The MD and DFT data are available at https://doi.org/10.24435/materialscloud:r1-e1. Source data are provided with this paper.

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

## Acknowledgements

Q.P. greatly acknowledges the financial support from the National Natural Science Foundation of China (52331003, 52288102 and 52171126), the Natural Science Foundation of Hebei Province (E2023203255 and C2022203003) and the Ministry of Education Yangtze River Scholar Professor Program (T2020124). G.Z. would like to express our gratitude to National Natural Science Foundation of China (52202374 and 52471050), Hebei Province high-level talent funding (HY2024050009) and Science Research Project of Hebei Education Department (BJK2024087). Z.W.S. is supported by the Singapore National Research Foundation (NRF Investigatorship NRF-NRFI09-0002) and the Agency for Science, Technology and Research (MTC Programmatic Fund M23L9b0052). In addition, we thank Prof. Liqiang Zhang from Yanshan University for his help with in situ TEM measurement. We also would like to extend our gratitude to Prof. Jean-Jacques Gaumet from Université de Lorraine for his language help.

## Author contributions

Q.P., G.Z. and J.W. conceived the project. G.Z., J.W., Y.S., T.N. and L.R. carried out the materials syntheses and structural characterizations. G.Z., J.W. and W.Y. conducted the electrochemical measurements. G.Z., J.W. and J.L. carried out computational investigation and provided theoretical analysis. G.Z. and J.W. wrote the manuscript with assistance from coauthors. Z.S. reviewed the paper. Q.P. was responsible for the overall direction of the project. All the other authors participated in preparing the manuscript and contributed to the discussion.

## Competing interests

The authors declare no competing interests.
