## [Peer Review File · Nature Communications]

REVIEWER COMMENTS

Reviewer #1 (Remarks to the Author):

The authors demonstrate a novel type of nanotwinned alloy capable of enabling ultrafast sodium deposition dynamics. Extensive experimental work and various methodologies have demonstrated the advantages of nanotwins, including long cycle life, high capacity retention, high specific capacity, high specific energy, and the absence of dendrite formation. The study elucidates the underlying mechanisms of uniform deposition in sodium metal anodes using nanotwinned alloys, providing novel and valuable insights for the development of dendrite-free and stable sodium metal anodes. Overall, the conclusions and data interpretation presented in this paper are robust, valid, and reliable, offering some significant contributions to the field of research. Its publication will provide important insights to researchers in the areas of alloys and energy storage. However, for the benefit of the readers, several issues need to be addressed before the manuscript can be accepted for publication in Nature Communications.

1. The authors are investigating nanotwinned alloys as collectors, but the preparation strategy for nanotwinned alloys should not be limited to current collector materials; it may also be applicable to alloyed electrode materials. To attract more interest from researchers, "collectors" should be removed from the title.
2. High-pressure processing appears to be a very challenging technique to implement and difficult to accomplish. Additionally, the authors should clearly articulate the advantages of using high-pressure methods.
3. If normal pressure solid solution combined with joule heating is used, how will the distribution of Si be affected? Will there be any performance improvements?
4. The authors created a sample containing 10 wt.% Si. What is the rationale for selecting a Si content of 10 wt.%?
5. In Fig. 1c, there is a significant difference in the twin fraction obtained from different processing strategies. Where do these values come from and how are they determined?
6. How does the grain size change before and after alloy processing?
7. There are errors in the abbreviations in Fig. 3; They should be HPJH.
8. The unit for the y-axis in Fig. 4b needs to be provided.
9. The authors should provide the detailed parameters of cycling measurements, especially for the SEM images in Fig. S26 that were obtained after cycling.
10. How was the cumulative capacity of the symmetrical cell calculated in Fig. 5?
11. Is the high-pressure treatment combined with joule heating only applicable to AlSi and CuAg systems?
12. The moives in the supplementary materials are blurry and high-resolution moives should be

provided.

Reviewer #2 (Remarks to the Author):

The Na metal battery, employing a nanotwinned alloy collector prepared by a high-pressure solid solution followed by joule-heating treatment, attains stable cycling over 5300 hours with a cumulative capacity of 13.25 Ah cm², representing the most stable collector in Na metal batteries reported to date, and it provides a potential strategy to enhance the next generation Na battery performance. The synthesis process is simple and applicative. The microstructure and twin-induced Na deposition mechanism are thoroughly assessed, and well-documented. The paper effectively details the in-situ reaction processes and the resulting products during discharge-charge process. The simulations or calculations are consistent with the experimental results. I believe this work will capture the interest of Nat. Commun's readership. However, the authors must clarify and carefully address the following issues or concerns before considering publication:

1. Why does the alloy subjected to high-pressure solid solution treatment combined with Joule heating exhibit such a large number of nanotwins? The authors should provide solid evidence and a thorough discussion to explain this phenomenon.
2. Twin fraction is closely related to the Na deposition, Meanwhile, the twin structure also affects the Na nucleate. Some simple twins and intersection twins are coexisted in the Si particles, if they play the same roles to induce the Na deposition.
3. What is the stability of the nanotwins? After prolonged electrochemical cycling, can the nanotwins in the current collector maintain their high twin density? Is there any potential loss during this process?
4. In Fig. 2 and Fig. S7, what was the rationale behind selecting the four electrolytes for observing sodium deposition morphology? Additionally, what was the reasoning for choosing 1 M NaPF₆ in DIGLYME for the subsequent battery tests?
5. In Fig. 4d, how were the two dashed lines that differentiate between dendritic, island-like, and planar deposition obtained? What is their practical significance?
6. The authors describe that the nanotwin collectors with low diffusion barriers guide the nucleation and growth of spherical Na, which subsequently forms a relatively flat Na deposition layer. How does the transition from spherical Na to a flat Na deposition layer occur?
7. In Figures 3 and 5, the time of in-situ experiments is different from the cell testing results, what are the main reasons.
8. The binding energy values are incorrectly labeled in Fig. S38.

“Nanotwinned Alloy Collectors Enable Ultrafast Sodium Deposition Dynamics”- Review

Feedback

I would like to appreciate the novel and innovative approach used by authors to address the challenge of sluggish sodium dynamics in sodium-metal batteries. The work is very well conceptualized and is strongly backed up with data representation – both experimental and computational. The manuscript presents a comprehensive study on utilizing the Silicon nano-twinned alloy collector to switch the deposition dynamics from diffusion-controlled to reaction-controlled, thus mitigating the issue of dendrite-formation. Demonstration of in-situ sodium deposition and in-depth analysis showcases a significant advancement in the field.

However, I have the following suggestions that I believe will enhance the clarity and impact of your work:

Comments-

1. In current collectors where maximizing electrical conductivity is the primary concern, alloying aluminium with silicon would reduce the electrical conductivity. Alloying is beneficial in case enhancement in mechanical properties like strength and ductility is important. Though a bit less favorable, nanotwins in Aluminium alone could be formed by severe plastic deformation or rapid solidification, without sacrificing electrical conductivity. Authors are suggested to elaborate on justification for selection of alloying route.
2. TEM image (Fig. 6 of supplementary resource) shows formation of silicon nanotwins. However, TEM focusses on much smaller area, limiting the field of view. Therefore, authors are recommended to examine relatively larger area to provide comprehensive overview of grain structure, orientation, and phase distribution through EBSD and perform CAM & GOS mapping.
3. Fig. 10 (f) cross-sectional SEM image shows bent substrate (Al current collector). Kindly provide a reason for this observation.
4. Fig. 21 – The coulombic efficiency Vs cycle number shows larger number of cycles for as-cast Al-Si as compared to HP Al-Si, which does not align with rest of the test results. Also, the data is quite scattered beyond 75th cycle in case of HP Al-Si system and 175th cycle for Al-Si sample. Please specify the reason for this. Authors are advised to indicate the experimental error or statistical dispersion of the results for different sets of repeated experiments. Also, provide coulombic efficiency plot for HPJH-AlSi system.
5. Figure 23- Can the voltage profile be provided for 80% DOD, which is usually the standard in performance metrics to allow for comparisons across different technologies.
6. How is the total active weight calculated? In sodiated or de-sodiated state?

7. It would be beneficial to include more detailed analysis and discussion regarding the underlying mechanisms driving the ultrafast sodium deposition observed. Cyclic voltammetry and differential plot (dQ/dV) could be utilized for the same.
8. "The microstructural morphologies and elemental compositions were observed through SEM (FEI Helios G4CX) at an accelerating voltage of 5 kV for SEM imaging and at 20 kV for EDS mapping"- Sodium has a low melting point of 97.8°C . In a high-vacuum SEM, the electron beam can cause localized heating of the sample. Since sodium has a low melting point, even modest beam currents or prolonged exposure may cause the sodium to melt or deform. How was the risk of surface bubbling/ melting managed at high voltage like 20 kV? Also, mention the dwell time applied.
9. Authors are advised to perform XPS analysis of the HPJH-AlSi/Na, HPA-AlSi/Na and Al/Na electrodes after cycling to understand the chemical nature of oxides formed and correlate it with cell performance.
10. Fig. 2- SEM micrographs show the images of morphology as a combination of substrate and electrolyte. Fig. 2c shows more granular, uniform and compact morphology than the one used for deposition (Fig. 2b). In this regard, why 1M NaClO_4 in DEC:EC as electrolyte was not preferred?
11. Fig. 22 c (supplementary material) shows stripping/plating stability of HP-AlSi system. The plot shows abrupt behavior at 100^{th} cycle. Please explain it. Also, provide the plot for HPJH-AlSi half cell.
12. The manuscript should not start with a symbol like "Na". Authors are advised to use name instead of symbol in the beginning. Also, a "conclusion section" at the end would be beneficial for readers.

Reply to the report of referee #1

Comment of Referee #1:

Reviewer 1

The authors demonstrate a novel type of nanotwinned alloy capable of enabling ultrafast sodium deposition dynamics. Extensive experimental work and various methodologies have demonstrated the advantages of nanotwins, including long cycle life, high capacity retention, high specific capacity, high specific energy, and the absence of dendrite formation. The study elucidates the underlying mechanisms of uniform deposition in sodium metal anodes using nanotwinned alloys, providing novel and valuable insights for the development of dendrite-free and stable sodium metal anodes. Overall, the conclusions and data interpretation presented in this paper are robust, valid, and reliable, offering some significant contributions to the field of research. Its publication will provide important insights to researchers in the areas of alloys and energy storage. However, for the benefit of the readers, several issues need to be addressed before the manuscript can be accepted for publication in Nature Communications.

Response: We thank this kind referee for the positive and kind comments on this manuscript. Our response follows each comment:

1) *The authors are investigating nanotwinned alloys as collectors, but the preparation strategy for nanotwinned alloys should not be limited to current collector materials; it may also be applicable to alloyed electrode materials. To attract more interest from researchers, "collectors" should be removed from the title.*

Response: We would like to thank the reviewer for his/her constructive suggestion. To broaden the scope of the study and appeal to a wider audience, we have removed the word 'collector' from the title. The revised title is: **A Nanotwinned-Alloy Strategy Enables Ultrafast Sodium Deposition Dynamics.**"

2) *High-pressure processing appears to be a very challenging technique to implement and difficult to accomplish. Additionally, the authors should clearly articulate the advantages of using high-pressure methods.*

Response: We express our gratitude for your insightful comment. While high-pressure processing can indeed be perceived as challenging, it has undergone significant advancements in recent years, making it a well-established and scalable technique [Nat. Commun., 2024, 15(1), 3932; Nature, 2014, 510(7504), 250-253]. With improvements in equipment and methodologies, it is now feasible to implement high-pressure synthesis on an industrial scale, with the ability to produce large-size alloys at a competitive cost. In our work, we have successfully synthesized large-sized alloys with a diameter of 56 mm and a

height of 20 mm (**Figure R1**), demonstrating the capability of the high-pressure process to produce substantial alloy specimens suitable for a wide range of applications.

More importantly, the main advantage of high-pressure techniques lies in their ability to significantly extend the solubility limits of alloying elements, leading to enhanced material properties that are difficult to achieve with conventional methods [*Scripta Mater.*, 2021, 201, 113970; *Corros. Sci.*, 2018, 143, 229-239]. In the Al-Si alloy system, where the solubility of Si under normal pressure is limited to just 1.6 wt.%. However, when synthesized at 675 °C under a pressure of 5 GPa, the solubility of Si can be increased to 10.4 wt.% (**Figure R2**). This considerable increase in solubility enhances the mechanical properties of the material, making it highly beneficial for applications requiring higher strength and stability. Moreover, this solubility extension through high-pressure methods is unique and cannot be easily replicated by other synthesis approaches, such as conventional melting or casting techniques. Therefore, the high-pressure process is not only feasible and scalable, but also offers significant advantages in expanding the solid solution capacity of the material, which is essential for optimizing the microstructure and improving the functional properties of advanced alloys.

Figure R1. Dimensions of HP-AlSi alloy.

Figure R2. The effect of pressure on the Al-Si alloy phase diagram.

3) If normal pressure solid solution combined with joule heating is used, how will the distribution of Si be affected? Will there be any performance improvements?

Response: Thank you for your valuable comment. We have conducted experiments using a normal pressure solid solution process combined with joule heating (NPJH). As a result, NPJH-AlSi alloys have a large amount of Si remaining undissolved in the Al substrate due to the limited solubility increase at normal pressure (**Figure R3a-c**). During Joule heating, only a small fraction of Si dissolves and subsequently precipitates as Si particles. Meanwhile, the precipitated Si particles are relatively small, about 20 nm, and no twins are observed inside the Si particles (**Figure R3d-f**). This evidence underscores the critical role of high-pressure solid solution process in promoting the formation of nanotwins, because it significantly increases the solubility of Si and promotes subsequent precipitation behavior of Si, facilitating the formation of numerous nanotwins in the alloy.

In terms of performance, we have conducted symmetric cell cycling tests on NPJH-AlSi alloys samples. The results indicated that NPJH-AlSi alloy collector could only cycle for 75 h at a current density of 5 mA cm^{-2} and a capacity of 5 mAh cm^{-2} , which is significantly lower than the performance of HPJH-AlSi (**Figure R3g-h**). This demonstrates that high-pressure treatment is essential for achieving superior electrochemical performance in these materials.

We have made the following changes in the revised manuscript:

*Furthermore, to demonstrate the superiority of the HPJH alloying strategy, we prepare pure Al coated with a Si layer and AlSi alloy treated with normal pressure solid solution combined with joule heating treatment for comparison. The weak bonding between the Si coating and the Al substrate causes detachment during deposition and stripping, resulting in the cycling performance of the Si-coated-Al symmetric cells being inferior (**Supplementary Fig. 33-35**). In addition, AlSi alloys with normal pressure solid solution combined with joule heating treatment have a large amount of Si remaining undissolved in the Al substrate due to the limited solubility increase at normal pressure (**Supplementary Fig. 36a-c**). During Joule heating, only a small fraction of Si dissolves and subsequently precipitates as Si particles. Meanwhile, the precipitated Si particles are relatively small, about 20 nm, and no twins are observed inside the Si particles (**Supplementary Fig. 36d-f**). As a result, the current collector is only able to 75 h at a current density of 5 mA cm^{-2} and a capacity of 5 mAh cm^{-2} , which is significantly lower than the performance of HPJH-AlSi. (**Supplementary Fig. 36g-h**). { Page 15, Line 21 }*

Figure R3. (a-c) SEM image of the AlSi alloy prepared by normal pressure solid solution process combined with joule heating (NPJH), and the corresponding element analysis spectrum. (d) TEM image of the NPJH-AlSi alloy. (e, f) HRTEM images of the NPJH-AlSi alloy. (g) Voltage profiles of the NPJH-AlSi alloy at 5 mA cm⁻², 5 mAh cm⁻². (h) Magnified view of selected cycle times between 50 and 75 h for the symmetric cell assembled with the NPJH-AlSi alloy.

NPJH-AlSi samples were solution treated at 500°C for 24 h, followed by an additional JH treatment at 800°C for 60 ms.

4) *The authors created a sample containing 10 wt.% Si. What is the rationale for selecting a Si content of 10 wt.%?*

Response: Thank you for your insightful question regarding the rationale behind selecting a Si content of 10 wt.%. Our selection of Si content is grounded in the thermodynamic properties of the Al-Si system, specifically the solid solubility limit of Si in Al under high-pressure solid solution conditions. According to the Al-Si phase diagram and previous research, Si has a maximum solid solubility of approximately 10.4 wt.% in Al at a high pressures of 5 GPa (**Figure R4**) [*J. Mater. Sci.* 1999, 34, 4149-4152]. To fully exploit this solubility and create an alloy with optimal properties, we chose a Si content of 10 wt.%.

This composition allows for the maximum incorporation of Si into the Al matrix during the high-pressure solid solution process. It is particularly important because achieving near-saturation of Si ensures that the subsequent Joule heating treatment has the highest possible thermodynamic driving force [*Adv. Energy Mater.*, 2022, 12(8), 2103505; *Adv. Energy Mater*, 2024, 14(5), 2302484; *Nati. Sci. Rev.*, 2024, 11(4), nwa033]. This driving force is critical for promoting the formation of nanotwinned Si structures within the Al matrix. These nanotwinned structures play a pivotal role in enhancing Na deposition dynamics and regulating the deposition behavior by transitioning from diffusion-controlled

to reaction-controlled processes, which directly influences the electrochemical performance of the Na metal batteries. Therefore, selecting a Si content of 10 wt.% not only maximizes the formation of these beneficial structures but also is consistent with the goal of enhancing Na deposition performance of the current collector.

Figure R4. The effect of pressure on the Al-Si alloy phase diagram.

5) In Fig. 1c, there is a significant difference in the twin fraction obtained from different processing strategies. Where do these values come from and how are they determined?

Response: Thank you for your insightful comment. The significant difference in twin fraction between the two processing strategies can be attributed to the distinct processing conditions used for each sample. In the case of HPJH-AlSi, a high-pressure solid solution process combined with Joule heat treatment was employed. This process induces a rapid change in the free energy of the supersaturated solid solution alloy post high-pressure treatment, leading to the instantaneous precipitation of solute Si [Adv. Energy Mater., 2022, 12(8), 2103505; Adv. Energy Mater, 2024, 14(5), 2302484]. As a result, the proportion of nanotwinned Si (NT-Si) in the HPJH-AlSi sample increased significantly. On the other hand, HPA-AlSi was processed through a high-pressure solid solution method followed by low-temperature aging treatment. This strategy leads to a much more gradual change in free energy, resulting in the slow precipitation of solute Si without the formation of nanotwins.

In addition, the twin fractions of 82.7% in HPJH-AlSi and 6.8% in HPA-AlSi were determined through TEM analysis of the Si morphology. The presence of stripe-like structures indicates the formation of nanotwins. The values reported are the averages obtained from three different TEM images for each sample, ensuring accuracy and reliability in the quantification of the twin fractions. These stripe-like features are characteristic of twin boundaries at the atomic level, where two regions of the crystal exhibit mirrored lattice symmetry across the twin plane. In TEM, these boundaries produce alternating dark and light stripes due to diffraction contrast from the differing orientations

of the twinned and untwinned regions. This contrast is a recognized indicator of nanotwins in crystalline materials.

6) *How does the grain size change before and after alloy processing?*

Response: Thank you for the comment regarding the grain size changes before and after alloy processing. We have added optical microscopy images of different Al-Si alloy samples, including Al, As-cast AlSi, HP-AlSi, HPA-AlSi, and HPJH-AlSi, alongside a comparison of grain sizes (**Figure R5**). Specifically, the grain sizes of Al, As-cast AlSi, HP-AlSi, HPA-AlSi, and HPJH-AlSi are 210, 92, 142, 153, and 157 μm , respectively. However, although there are differences in grain size between the samples, it must be emphasized that the focus of our study is not on the impact of grain size variation, but rather on the distribution of solute Si and the role of nanotwinned Si in enhancing electrochemical performance. We find that the uniform nanotwins play a more significant role in promoting Na deposition and contributing to the reaction-controlled deposition mechanism than the effect of grain size variation. While grain size may affect mechanical properties of other systems, in this case, it does not have a direct impact on the observed electrochemical behaviour. Therefore, the variations in grain size observed among the different samples (Al, As-cast AlSi, HP-AlSi, HPA-AlSi, and HPJH-AlSi) are not a key factor in our analysis and do not affect the conclusions drawn from the study.

We have made the following changes in the revised manuscript:

*The grain sizes of the above samples with five different alloy treatment processes have been shown in **Supplementary Fig. 4**. {Page 6, Line 8}*

Figure R5. Grain size changes of different samples. Optical images of (a) Al, (b) As-cast-AlSi, (c) HP-AlSi, (d) HPA-AlSi, and (e) HPJH-AlSi. f. Comparison of grain sizes of different samples.

7) There are errors in the abbreviations in Fig. 3; They should be HPJH.

Response: We thank the reviewer for pointing out the errors in the abbreviations in Fig. 3. We have carefully reviewed the manuscript and made the necessary corrections. The abbreviations have been updated to 'HPJH' as appropriate. The revised Figure 3 (**Figure R6**) is included below for your reference.

Figure R6. In-situ analysis of Na deposition. (a) Schematic of the electrochemical deposition setup. (b-g) Time lapse TEM images of Na metal deposition on HPJH-AISi collector. Scale bar, 5 μm . (h) Time-resolved evolution of Na size as a function of time during Na deposition on different substrates. i-n The temporal evolution of Na deposition on HPJH-AISi obtained with MD simulations.

8) The unit for the y-axis in Fig. 4b needs to be provided.

Response: We thank the reviewer for highlighting the omission of the unit for the y-axis in Fig. 4b (**Figure R7**). We have made the necessary corrections in the manuscript, and the unit for MSD on the y-axis has been added as A^2 . The revised figure is included below for reference.

Figure R7. MSD simulation of Na on different collectors.

9) The authors should provide the detailed parameters of cycling measurements, especially for the SEM images in Fig. S26 that were obtained after cycling.

Response: We appreciate the reviewer's suggestion to provide more detailed parameters for the cycling measurements, especially for the SEM images in Fig. S26 obtained after cycling. We have added this information to the manuscript. The images were taken after the electrode was cycled at a current density of 5 mA cm^{-2} and a capacity of 5 mAh cm^{-2} for 10 cycles (**Figure R8**). The updated figure caption has also been revised to reflect these details.

We have made the following changes in the revised manuscript:

*Additionally, after 10 cycles at a current density of 5 mA cm^{-2} and a capacity of 5 mAh cm^{-2} , the HPJH-AlSi current collector still retains a dense distribution of nanotwinned Si and exhibits a well-preserved Na plating/stripping morphology (**Supplementary Fig. 30, 31**). Comparatively, the pronounced dendritic and nonuniform distribution of Na on both the Al and HPA-AlSi collectors exacerbates parasitic reactions (**Supplementary Fig. 31**). {Page 15, Line 10}*

Figure R8. SEM images (a-c) and optical images (d-f) of HPJH-AlSi/Na (a, d), HPA-AlSi/Na (b, e), and Al/Na (c, f) electrodes after cycling at a current density of 5 mA cm^{-2} and a capacity of 5 mAh cm^{-2} for 10 cycles.

10) How was the cumulative capacity of the symmetrical cell calculated in Fig. 5?

Response: We thank the reviewer for the question regarding the calculation of cumulative capacity in Fig. 5. The cumulative capacity of the symmetrical cell was calculated using the following formula [Energy Environ. Sci., 2021, 14(1), 382-395]:

Cumulative capacity = Capacity per cycle \times Cycle number

with the units expressed in Ah cm⁻². This formula has been added to the supplementary materials, following the cumulative capacity table (Supplementary Table 2) for clarity.

We have also made the following modifications to the supporting manuscript:

The cumulative capacity of the symmetrical cell was calculated using the following formula:

Cumulative capacity = Capacity per cycle \times Cycle number, in units of Ah cm⁻² {Supporting information Page 52, Line 1}

11) Is the high-pressure treatment combined with joule heating only applicable to AlSi and CuAg systems?

Response: We appreciate the reviewer's insightful question regarding the applicability of high-pressure treatment combined with Joule heating to other alloy systems. Currently, we believe this method is primarily suitable for alloys without intermediate phases and with a significant solubility change under high pressure. These two conditions are essential for ensuring the formation of a supersaturated solid solution during high-pressure treatment and for achieving a substantial free energy change during subsequent joule heating, which enables rapid solute precipitation and the formation of nanotwins.

Based on these criteria, we hypothesize that the process could also be applicable to binary alloys such as Al-Ge (**Figure R9a**). The phase diagrams of these alloys are similar to those of Al-Si and Cu-Ag (**Figure R9b, c**), suggesting the potential for a similar behavior under high-pressure solid solution followed by joule-heating treatment.

Figure R9. Phase diagram of (a) AlGe, (b) AlSi, (c) CuAg.

12) The moives in the supplementary materials are blurry and high-resolution moives should be provided.

Response: We thank the reviewer for pointing out the issue with the quality of the movies in the supplementary materials. We have now replaced the previously uploaded files with high-resolution versions to ensure clarity and better visualization.

Reply to the report of referee #2

Comment of Referee #2:

Reviewer 2

The Na metal battery, employing a nanotwinned alloy collector prepared by a high-pressure solid solution followed by joule-heating treatment, attains stable cycling over 5300 hours with a cumulative capacity of 13.25 Ah cm², representing the most stable collector in Na metal batteries reported to date, and it provides a potential strategy to enhance the next generation Na battery performance. The synthesis process is simple and applicative. The microstructure and twin-induced Na deposition mechanism are thoroughly assessed, and well-documented. The paper effectively details the in-situ reaction processes and the resulting products during discharge-charge process. The simulations or calculations are consistent with the experimental results. I believe this work will capture the interest of Nat. Commun's readership. However, the authors must clarify and carefully address the following issues or concerns before considering publication:

Response: We thank the referee for the positive and helpful remarks. Our response follows each comment below.

1) Why does the alloy subjected to high-pressure solid solution treatment combined with Joule heating exhibit such a large number of nanotwins? The authors should provide solid evidence and a thorough discussion to explain this phenomenon.

Response: We are grateful for your thoughtful comment. The formation of a large number of nanotwins in the alloy subjected to high-pressure solid solution treatment combined with Joule heating can be attributed to the enhanced solubility of solute Si under high pressure and the rapid precipitation during Joule heating. Specifically, during high-pressure solid solution treatment, the solubility of Si atoms in the Al matrix increases significantly. Under extreme pressure conditions, such as 5 GPa, the solubility of Si in Al can rise to 10 wt.% (**Figure R10**). This results in the complete dissolution of the eutectic Si, which forms under normal pressure conditions, into the Al matrix, creating a metastable supersaturated solid solution [*J. Mater. Sci.* 1999, 34, 4149-4152]. This process is crucial for achieving a high fraction of nanotwins, as it provides the necessary preconditions for their subsequent formation. To demonstrate the effect of high-pressure solid solution treatment, we conducted experiments using a normal pressure solid solution process combined with Joule heating (NPJH). The results revealed that, due to the limited increase in solubility under normal pressure, a large amount of Si remained undissolved in the Al matrix. Only a small fraction of Si was dissolved and subsequently precipitated as Si particles during the Joule

heating treatment (**Figure R11a-c**). Meanwhile, the precipitated Si particles are relatively small, about 20 nm, and no twins are observed inside the Si particles (**Figure 11d-f**).

This evidence underscores the critical role of high-pressure solid solution process in promoting the formation of nanotwins, because it significantly increases the solubility of Si and promotes subsequent precipitation behavior of Si, facilitating the formation of numerous nanotwins in the alloy.

Furthermore, different aging strategies applied to the alloy after high-pressure solid solution treatment yield significantly different results. When combined with low-temperature aging (HPA-AlSi), the slow precipitation of Si solute results in a lower fraction (6.8%) of nanotwins in the HPA-AlSi (**Figure R12**). In contrast, during Joule heating, the rapid change in free energy of the metastable supersaturated solid solution causes the transient non-equilibrium relaxation precipitation of solute Si [*Adv. Energy Mater.*, 2022, 12(8), 2103505; *Adv. Energy Mater.*, 2024, 14(5), 2302484; *J. Mater. Chem. A*, 2024, 12, 23712-23722], significantly increasing the proportion of nanotwinned Si in the HPJH-AlSi alloy collector, reaching a high value of 82.7% (**Figure R13**).

In summary, the formation of a large number of nanotwins is primarily driven by the combination of high-pressure solid solution treatment and rapid Joule heating. The former enhances solubility and forms a metastable alloy matrix, while the latter induces the appearance of twin structure through transient non-equilibrium relaxation precipitation. These combined effects explain the high density of nanotwins observed in the alloy collectors.

Figure R10. The effect of pressure on the Al-Si alloy phase diagram

Figure R11. (a-c) SEM image of the AlSi alloy prepared by conventional solid solution process combined with Joule heat, and the corresponding element analysis spectrum. (d) TEM image of the AlSi alloy.(e, f) Atomic resolution HAADF image of the AlSi alloy.

Figure R12. TEM image of the HPA-AlSi and the inset shows the HRTEM image of Si nanocrystals

Figure R13. (a) TEM image of the HPJH-AlSi. Scale bar, 200 nm. (b) Fraction of NT-Si supported on HPJH-AlSi and HPA-AlSi statistically counted from TEM images.

2) Twin fraction is closely related to the Na deposition, Meanwhile, the twin structure also affects the Na nucleate. Some simple twins and intersection twins are coexisted in the Si particles, if they play the same roles to induce the Na deposition.

Response: We appreciate your valuable comment. The twin fraction indeed plays a critical role during Na deposition. Specifically, the HPJH-AlSi alloy subjected to high-pressure solid solution combined with joule heat treatment contains up to 82.7% of nanotwinned Si, which significantly improves the deposition dynamics through its excellent Na diffusion rate. This results in a Damköhler number far below 1, successfully shifting the Na deposition process from diffusion-controlled to reaction-controlled, ultimately leading to spherical Na deposition (**Figure R14a, c**). In contrast, HPA-AlSi alloys with low twin fractions lack the sufficient twin fraction needed to enhance Na deposition dynamics, resulting in a Damköhler number of approximately 1. This suggests that the rate of surface self-diffusion is very close to the rate of electrochemical reactions, and that the morphology of the Na deposits is primarily columnar-island-like (**Figure R14b, c**).

Moreover, both simple twins and intersection twins are twinned along the [011] axis and the {111} plane. Although their orientations may differ in orientation, their crystal structures are fundamentally identical, and this observation has been reported in previous literature [*Mater. Sci. Eng. A, 2018, 712, 757-764; Mater. Sci. Eng. A, 2019, 751, 303-310*]. In addition, these twin structures can effectively provide stable nucleation sites for Na and thus induce uniform Na deposition. In other words, both simple and intersection twins are able to promote the deposition behaviour of Na under the same physical mechanism, which ultimately shows a consistent effect.

Figure R14. The diffusion pathway and calculated energy profile of Na along the diffusion path of Na on the NT-Si (a) and Si (b). c Deposition morphology phase map as a function of Damköhler number.

3) What is the stability of the nanotwins? After prolonged electrochemical cycling, can the nanotwins in the current collector maintain their high twin density? Is there any potential loss during this process?

Response: Thank you for your insightful comments. To address your concerns, we performed TEM characterization on HPJH-AlSi after electrochemical testing to evaluate the stability of the nanotwinned Si structure (**Figure R15**). Our findings indicate that the high density of nanotwins is largely preserved, even after 10 cycles. Such high retention is likely due to the mechanical strength provided by the nanotwins, combined with their atomic-level stability, helps to mitigate any significant structural changes [*Mater.Design*, 2020, 192, 108752; *Acta Mater.*, 2019, 165, 142-152; *Sci. Adv.*, 2021, 7(27), eabg5113]. Thus, the nanotwins maintain their density and integrity under cycling conditions. In summary, the nanotwinned Si in the HPJH-AlSi alloy current collectors are able to maintain high density with negligible loss throughout the electrochemical process. This stability is key to the improved Na deposition dynamics and enhanced electrochemical performance.

We have made the following modifications to the manuscript:

Additionally, after 10 cycles at a current density of 5 mA cm⁻² and a capacity of 5 mAh cm⁻², the HPJH-AlSi current collector still retains a dense distribution of nanotwinned Si and exhibits a well-preserved Na plating/stripping morphology (Supplementary Fig. 30). {Page 15, Line 10}

Figure R15. (a) TEM characterization of HPJH-AlSi after electrochemical testing. (b-g) HRTEM images in the HPJH-AlSi with nanotwin.

4) In Fig. 2 and Fig. S7, what was the rationale behind selecting the four electrolytes for observing sodium deposition morphology? Additionally, what was the reasoning for choosing 1 M NaPF₆ in DIGLYME for the subsequent battery tests?

Response: We appreciate the reviewers' insightful comments. For the selection of electrolytes, we have provided detailed explanations below:

First, we chose four electrolytes (1 M NaPF₆ in diglyme =100 vol%, 2.1 M NaFSI in DME:BTFE = 1:2 mol%, 1 M NaClO₄ in DEC:EC=1:1 vol% with 5% FEC, and 1 M NaClO₄ in PC=100 vol% with 5% FEC) to comprehensively evaluate the Na deposition behavior on the different collector (HPJH-AISi, HPA-AISi, and Al). Specifically, we selected two solvent systems: ether-based (1 M NaPF₆ in diglyme =100 vol%, 2.1 M NaFSI in DME:BTFE = 1:2 mol%) and ester-based (1 M NaClO₄ in DEC:EC=1:1 vol% with 5% FEC, 1 M NaClO₄ in PC=100 vol% with 5% FEC), both of which are widely used in Na battery research. Each system has its own advantages [*ACS Central Sci.*, 2015, 1(8), 449-455; *ACS Energy Lett.*, 2018, 3(2), 315-321; *ChemElectroChem*, 2016, 3(11), 1856-1867; *ACS Sustainable Chem. Eng.*, 2017, 5(9), 8269-8276; *J. Energy Storage*, 2023, 72, 108781]. Ether-based electrolytes typically exhibit better Na ion conductivity and a lower tendency for Na dendrite formation, while ester-based electrolytes can demonstrate enhanced stability through additive tuning in certain cases [*Chem. Soc. Rev.*, 2022, 51(11), 4484-4536; *InfoMat*, 2019, 1(3), 376-389]. By selecting a diverse range of electrolytes, we were able to systematically evaluate whether the electrolyte choice affects Na deposition behavior. The results showed that the modified nanotwinned collector-HPJH-AISi consistently exhibited spherical Na deposition, regardless of the electrolyte, whereas dendritic formation was observed on the pure Al collector. Therefore, we have conclusively demonstrated that the deposition behaviour is mainly influenced by the collector rather than the composition of the electrolyte. This highlights the versatility and wide applicability of the improved nanotwinned alloy collector in various electrolyte systems.

In subsequent battery tests, we selected the 1 M NaPF₆ in diglyme electrolyte system based on preliminary findings, which demonstrated its superior Coulombic efficiency and stable long-term cycling performance [*ACS Central Sci.*, 2015, 1(8), 449-455]. Specifically, the 1 M NaPF₆ in diglyme system exhibits high Na ion conductivity, facilitating rapid ion migration during charge and discharge cycles [*ACS Appl. Energy Mater.*, 2018, 1(6), 2671-2680]. Efficient Na ion transport reduces internal resistance, enhancing both the rate performance and overall electrochemical efficiency. Moreover, this electrolyte offers excellent chemical and electrochemical stability, which helps prevent undesirable side reactions with deposited Na, thus mitigating performance degradation over time [*ACS Appl. Mater. Interfaces*, 2019, 11(36), 32844-32855]. Additionally, its ability to maintain low polarization during extended cycling makes it particularly well-suited for Na-metal battery

applications that demand long life and high reliability [ACS Central Sci., 2015, 1(8), 449-455]. For these reasons, 1 M NaPF₆ in diglyme was chosen for further testing due to its high ionic conductivity, chemical stability, low polarization, and excellent long-term cycling performance.

5) In Fig. 4d, how were the two dashed lines that differentiate between dendritic, island-like, and planar deposition obtained? What is their practical significance?

Response: We thank the reviewer for their insightful question regarding the origin and significance of the dashed lines in Fig. 4d. The dashed lines are derived from the isocontours of the Damköhler number (Da), which is the ratio of diffusion rate to reaction rate (**Figure R16**) [ACS Appl. Mater. Inter., 2018, 10(31), 26320-26327]. In this plot, the x-axis represents the diffusion rate, while the y-axis represents the reaction rate, and the dashed lines correspond to Da values of 10⁻² and 10², respectively.

In chemical reaction engineering, the Damköhler number is used to describe the relative importance of diffusion and reaction rates [ACS Energy Lett., 2018, 4(2), 375-376]. When **Da** \ll 1, diffusion dominates, meaning the reaction rate is limited by how quickly species can diffuse. This corresponds to the 'planar' region in the figure, where deposition occurs uniformly across the surface. In contrast, when **Da** \gg 1, the reaction rate is much faster than the diffusion rate, leading to uneven deposition, often resulting in dendritic growth, as observed in the 'dendritic' region. The 'island-like' region represents an intermediate state where diffusion and reaction rates are more balanced, leading to deposition in island-like forms.

Thus, the dashed lines mark critical Damköhler number values, delineating the transition between different deposition morphologies as the balance between reaction and diffusion shifts.

Figure R16. Deposition morphology phase map as a function of Damköhler number.

6) The authors describe that the nanotwin collectors with low diffusion barriers guide the nucleation and growth of spherical Na, which subsequently forms a relatively flat Na deposition layer. How does the transition from spherical Na to a flat Na deposition layer occur?

Response: We thank the reviewer for the insightful question regarding the transition from spherical Na to a flat Na deposition layer. This transition is a process of Na sphere growth and coalescence, which we have confirmed through MD simulations. The MD simulations show that between 0 to 600 ps, Na atoms gradually form spherical shapes, with the spherical characteristics becoming more pronounced by 800 ps. Eventually, the spherical Na structures begin to merge, and by 1000 ps, coalescence occurs, leading to a flatter Na deposition layer (**Figure R17**).

In addition, this coalescence process is further supported by our experimental observations using SEM testing. At a low deposition capacity of 0.1 mAh cm^{-2} , the Na deposition exhibits distinct spherical morphology, confirming the initial nucleation and growth of Na in the form of individual spheres. As the deposition capacity increases, particularly at 7 mAh cm^{-2} , these Na spheres coalesce to form a uniform, dense Na layer, as shown by the more compact and flatter deposition morphology in the SEM images (**Figure R18**). These findings collectively indicate that the low diffusion barriers provided by the nanotwinned collectors facilitate the smooth transition from spherical Na nucleation to a flat and stable Na deposition layer, contributing to enhanced performance.

Figure R17. (a-f) The temporal evolution of Na deposition on HPJH-AlSi obtained with MD simulations.

Figure R18. Top view SEM images of Na deposited on HPJH-AlSi at (a) 0.1 mAh cm⁻² (b) 7 mAh cm⁻² capacities.

7) In Figures 3 and 5, the time of in-situ experiments is different from the cell testing results, what are the main reasons.

Response: We appreciate the reviewer's question regarding the difference in time scales between the in-situ experiments in Figure 3 and the cell testing results in Figure 5. The objective of the in-situ deposition experiments in Figure 3 is to observe how the material forms or changes over a short period, providing real-time insight into the deposition dynamics (**Figure R19**). In contrast, the goal of the cell testing in Figure 5 is to evaluate the material's performance under practical conditions, which requires extended testing time (**Figure R20**).

Additionally, the environmental factors during deposition may have a significant impact on the results, and these conditions need to remain stable over shorter durations for the in-situ analysis. Meanwhile, the cell testing focuses on tracking the material's behavior over multiple cycles to ensure comprehensive performance evaluation over longer durations.

Figure R19. *In-situ* analysis of Na deposition. (a) Schematic of the electrochemical deposition setup. (b-g) Time lapse TEM images of Na metal deposition on HPJH-AlSi collector. Scale bar, 5 μm. h Time-resolved evolution of Na size as a function of time during Na deposition on different substrates. (i-n) The temporal evolution of Na deposition on HPJH-AlSi obtained with MD simulations.

Figure R20. Asymmetric and symmetric cell performances. (a) Coulombic efficiencies of Na plating/stripping on different collectors at current densities of 3 mA cm^{-2} and areal capacities of 1 mAh cm^{-2} . (b) Voltage-capacity profiles of HPJH-AISi at 3 mA cm^{-2} and 1 mAh cm^{-2} . (c) Voltage profiles of symmetric cells assembled with different collectors at 5 mA cm^{-2} and 5 mAh cm^{-2} . (d) Magnified view of selected cycle times between 5265 and 5300 h for the symmetric cell assembled with HPJH-AISi. (e) A comparative analysis of cumulative capacity and cycling time in the recent demonstration of Na symmetric cells with our work. Detailed information for each demonstration are provided in Supplementary Table 2. (f) Radar plots of the five main electrochemical properties of different collectors.

8) *The binding energy values are incorrectly labeled in Fig. S38.*

Response: We thank the reviewer for pointing out the incorrect labeling of the binding energy values in Fig. S38 (**Figure R21**). We have corrected the errors, and the updated figure is now provided below. Specifically, the binding energies of NT-Ag(111), Ag(111), and Cu(111) with Na atoms are -1.78 eV , -1.43 eV , and -0.93 eV , respectively.

Figure R21. Calculation of the binding energies of different crystal facets with Na atoms.

Reply to the report of referee #3

Comment of Referee #3:

Reviewer 3

I would like to appreciate the novel and innovative approach used by authors to address the challenge of sluggish sodium dynamics in sodium-metal batteries. The work is very well conceptualized and is strongly backed up with data representation – both experimental and computational. The manuscript presents a comprehensive study on utilizing the Silicon nano-twined alloy collector to switch the deposition dynamics from diffusion-controlled to reaction-controlled, thus mitigating the issue of dendrite-formation. Demonstration of in-situ sodium deposition and in-depth analysis showcases a significant advancement in the field.

However, I have the following suggestions that I believe will enhance the clarity and impact of your work:

Response: We are grateful for the reviewer critical evaluation and we hope that the revised version has answered all his or her comments.

1) In current collectors where maximizing electrical conductivity is the primary concern, alloying aluminium with silicon would reduce the electrical conductivity. Alloying is beneficial in case enhancement in mechanical properties like strength and ductility is important. Though a bit less favorable, nanotwins in Aluminium alone could be formed by severe plastic deformation or rapid solidification, without sacrificing electrical conductivity. Authors are suggested to elaborate on justification for selection of alloying route.

Response: Thank you for your insightful suggestions. We appreciate the opportunity to further clarify the rationale behind our alloying approach and provide additional data on electrical conductivity and mechanical properties to address your concerns.

First, alloying Al with Si indeed reduces electrical conductivity. However, our supplementary experimental results indicate that the electrical conductivity of the HPJH-AlSi alloy current collector is 2.77×10^6 S/m, compared to 8.94×10^6 S/m for pure Al (**Figure R22a**). While there is a decrease, the conductivity of the HPJH-AlSi alloy collector remains within the same order of magnitude as pure Al, which is sufficient for use in Na metal batteries. Regarding mechanical properties, the HPJH-AlSi alloy exhibits significant improvements over pure Al (**Figure R22b and Table R1**). Specifically, the HPJH-AlSi alloy has a yield strength of 344 MPa, a tensile strength of 456 MPa, and an elongation rate of 17%, while pure Al has a yield strength of 45 MPa, a tensile strength of 72 MPa, and an elongation rate of 37%. These enhanced mechanical properties are critical

for maintaining the structural integrity of the current collector during battery cycling, particularly during repeated Na deposition and stripping.

Moreover, as you say, nanotwin structures can be formed in Al through severe plastic deformation or rapid solidification without sacrificing electrical conductivity. However, in our study, the use of high-pressure solid solution combined with joule heating treatment allows for the formation of a large number of uniformly distributed nanotwinned Si structures within the Al matrix, which offers unique advantages in regulating Na deposition. Specifically, by introducing Si into the Al matrix, especially in the form of nanotwinned structures, the Na diffusion rate is significantly enhanced, resulting in a Damköhler number much less than 1 (**Figure R23**). This effectively shifts the deposition dynamics from diffusion-controlled to reaction-controlled, leading to more uniform Na deposition, as demonstrated in our in-situ Na deposition experiments (**Figure R24, 25**).

We have made the following modifications to the manuscript:

To clearly understand the intrinsic properties of the current collector before and after alloying, we conducted tests on the electrical conductivity and mechanical properties of both pure Al and HPJH-AlSi alloy (Supplementary Fig. 43 and Supplementary Table 6). Compared with pure Al, the electrical conductivity of the HPJH-AlSi alloy can remain the same order of magnitude (Supplementary Fig. 43a), indicating that its application in Na metal batteries is not significantly compromised. More importantly, the HPJH-AlSi alloy shows substantial improvements in mechanical properties in contrast to pure Al (Supplementary Fig. 43b). These enhancements are crucial for maintaining the structural integrity of the current collector during battery cycling, particularly under the repeated processes of Na deposition and stripping. {Page 18, Line 4}

We have also made the following modifications to the supporting information:

Regarding electrical conductivity tests: *The resistivity of the material is derived using the van der Pauw method (vdp). This four-probe method is applied to small, flat samples with four terminals and uniform thickness. A current is applied through two terminals of the sample, and the voltage drop is measured across the opposite two terminals. The specific formula is as follows:*

$$\begin{aligned}
 Q_A &= \left(\frac{R_{12,43}^+ - R_{12,43}^-}{R_{23,14}^+ - R_{23,14}^-} \right) = \left(\frac{V_{12,43}^+ - V_{12,43}^-}{I_{12}^+ - I_{12}^-} \right) \left(\frac{I_{23}^+ - I_{23}^-}{V_{23,14}^+ - V_{23,14}^-} \right) \\
 Q_B &= \left(\frac{R_{34,21}^+ - R_{34,21}^-}{R_{41,32}^+ - R_{41,32}^-} \right) = \left(\frac{V_{34,21}^+ - V_{34,21}^-}{I_{34}^+ - I_{34}^-} \right) \left(\frac{I_{41}^+ - I_{41}^-}{V_{41,32}^+ - V_{41,32}^-} \right) \\
 \frac{Q-1}{Q+1} &= \frac{f}{\ln 2} \cos h^{-1} \left\{ \frac{1}{2} \exp \left[\frac{\ln 2}{f} \right] \right\} \\
 R_A &= \frac{\pi \cdot f_A}{\ln 2} \left\{ \frac{V_{12,43}^+ - V_{12,43}^- + V_{23,14}^+ - V_{23,14}^-}{I_{12}^+ - I_{12}^- + I_{23}^+ - I_{23}^-} \right\} \\
 R_B &= \frac{\pi \cdot f_B}{\ln 2} \left\{ \frac{V_{34,21}^+ - V_{34,21}^- + V_{41,32}^+ - V_{41,32}^-}{I_{34}^+ - I_{34}^- + I_{41}^+ - I_{41}^-} \right\}
 \end{aligned}$$

$$\rho_A = R_A \cdot t$$

$$\rho_B = R_B \cdot t$$

$$\rho_{av} = \frac{\rho_A + \rho_B}{2}$$

Here, R is the sample resistance, A is the test current, V is the voltage, and ρ is the resistivity. t is the thickness of the sample and f is the geometric factor of the symmetry of the sample which is related to the two resistance ratios Q .

Finally, the reciprocal of the calculated resistivity is taken to obtain the electrical conductivity. {Supporting information, Page 44, Line 3}

Regarding mechanical performance tests: Dog-bone tensile test samples with a gauge of 8 mm were used in tensile tests. The tensile tests were performed at room temperature at an initial strain rate of $1.0 \times 10^{-3} \text{ s}^{-1}$. The value was averaged over at least three measurements. {Supporting information, Page 45, Line 7}

Figure R22. (a) electrical conductivity and (b) mechanical properties of Al and HPJH-AlSi.

Table R1. Summary of electrical conductivity and mechanical properties of Al and HPJH-AlSi.

Samples	Electrical conductivity (S/m)	Yield strength (MPa)	Tensile strength (MPa)	Elongation rate (%)
Pure Al	8.94×10^6	45	72	37
HPJH-AlSi	2.77×10^6	344	456	17

Figure R23. Deposition morphology phase map as a function of Damköhler number.

Figure R24. (a-f) Time lapse TEM images of Na metal deposition on HPJH-AISi collector. Scale bar, 5 μm .

Figure R25. (a-f) The temporal evolution of Na deposition on HPJH-AISi obtained with MD simulations.

2) TEM image (Fig. 6 of supplementary resource) shows formation of silicon nanotwins. However, TEM focusses on much smaller area, limiting the field of view. Therefore, authors are recommended to examine relatively larger area to provide comprehensive overview of grain structure, orientation, and phase distribution through EBSD and perform CAM & GOS mapping.

Response: We sincerely appreciate the reviewer's constructive feedback. In response to the suggestion to examine a larger area, we have supplemented our study with both optical microscopy observations and electron backscatter diffraction (EBSD) analysis. Specifically, the optical microscopy images confirmed that the HPJH-ALSi sample consists of equiaxed grains with an average grain size of approximately 157 μm (**Figure R26**). Furthermore, in order to provide a comprehensive overview of the grain structure, orientation, and phase distribution, we conducted EBSD analysis on the HPJH-ALSi sample, along with Kernel Average Misorientation (KAM) and Grain Orientation Spread (GOS) mapping. According to the EBSD inverse pole figure (IPF) map shown in **Figure R27a**, it is indicated that the samples are mainly composed of equiaxed grains, with uniform size distribution and no preferential orientation.

In addition to the IPF analysis, we performed GOS mapping, as shown in **Figure R27b**. The GOS map provides insight into the internal misorientation of each grain, helping differentiate between recrystallized and deformed grains. As shown in **Figure R27c**, grains with GOS values greater than 2° are classified as deformed, while those with values below 2° are considered recrystallized [*J. Magnes. Alloy*, 2024, 12, 2793-2811; *Materials*, 2022, 15(19), 6769]. The analysis revealed that 97.1% of the grains in the scanned region had a GOS value of less than 2° , indicating that the majority of the grains had undergone recrystallization.

To further corroborate these findings, we performed KAM mapping, which highlights local misorientation and provides information about dislocation density and plastic deformation. **Figure R27d** shows the KAM distribution, where low KAM values are observed across most grains and grain boundaries, suggesting minimal plastic deformation [*Mater. Sci. Eng. A*, 2021, 807, 140821; *Ultramicroscopy*, 2018, 184, 156-163]. The average intra-granular misorientation (θ_{KAM}) is calculated to be approximately 0.2° , further supporting the conclusion that the grains are largely free of significant deformation.

We have made the following modifications to the manuscript:

*The grain sizes of the above samples with five different alloy treatment processes have been shown in **Supplementary Fig. 4**. {Page 6, Line 8}*

However, based on electron backscatter diffraction (EBSD) analyses, the HPJH-ALSi alloy primarily comprises equiaxed grains with uniform size distribution, and there is no

preferential orientation among neighboring grains (**Supplementary Fig. 8**). This implies that no significant deformation occurs in the matrix grains. {Page 7, Line 12}

We have also made the following modifications to the supporting information:

According to the data of the EBSD inverse pole figure (IPF) shown in **Supplementary Fig. 8a**, it is indicated that the samples are mainly composed of equiaxed grains, with uniform size distribution and no preferential orientation. In addition to the IPF analysis, we performed GOS mapping, as shown in **Supplementary Fig. 8b**. The GOS map provides insight into the internal misorientation of each grain. As shown in **Supplementary Fig. 8c**, grains with GOS values greater than 2° are classified as deformed, while those with values below 2° are considered recrystallized. The analysis revealed that 97.1% of the grains in the scanned region had a GOS value of less than 2° , indicating that the majority of the grains had undergone recrystallization. To further corroborate these findings, we performed KAM mapping, which highlights local misorientation and provides information about dislocation density and plastic deformation. **Supplementary Fig. 8d** shows that the KAM distribution, where low KAM values are observed across most grains and grain boundaries, suggesting minimal plastic deformation. The average intra-granular misorientation (θ_{KAM}) was calculated to be approximately 0.2° , further supporting the conclusion that the grains are largely free of significant deformation. {Supporting information, Page 9, Line 3}

Figure R26. Grain size changes of different samples. Optical images of (a) Al, (b) As-cast-AlSi, (c) HP-AlSi, (d) HPA-AlSi, and (e) HPJH-AlSi. f. Comparison of grain sizes of different samples.

Figure R27. (a) EBSD IPF mapping of HPJH-AlSi alloy. (b) Grain orientation spread map. (c) GOS distribution. (d) Kernel average misorientation.

3) Fig. 10 (f) cross-sectional SEM image shows bent substrate (Al current collector). Kindly provide a reason for this observation.

Response: We are grateful for your thoughtful comment. The observed bending of the substrate (Al current collector) in Fig. 10 (f) is attributed to stress introduced during the cutting of the electrode. This factor may have caused mechanical deformation of the current collector. To provide a clearer illustration and better observe the Na deposition morphology on the Al current collector, we have re-prepared the sample and conducted additional SEM imaging. The updated cross-sectional SEM image is provided below (**Figure R28, Renewed Supplementary Fig. 12 f**). In this new image, it is evident that the Na deposition on the Al current collector forms a layer 35.71 μm thick and exhibits significant dendritic morphology.

Figure R28. Cross-section SEM images of Na deposited on Al under the current density of 2 mA cm^{-2} and capacities of 2 mAh cm^{-2} .

4) Fig. 21 – The coulombic efficiency Vs cycle number shows larger number of cycles for as-cast Al-Si as compared to HP Al-Si, which does not align with rest of the test results. Also, the data is quite scattered beyond 75th cycle in case of HP Al-Si system and 175th cycle for Al-Si sample. Please specify the reason for this. Authors are advised to indicate the experimental error or statistical dispersion of the results for different sets of repeated experiments. Also, provide coulombic efficiency plot for HPJH-AlSi system.

Response: We are grateful for your thoughtful comment. Thank you to the reviewers for their valuable comments on our work. We have carefully reviewed the feedback and made corresponding revisions to the manuscript. Below is a detailed response addressing each of the reviewers' comments:

First, we compared the performance of five samples: HPJH-AlSi, HPA-AlSi, As-cast-AlSi, HP-AlSi, and Al. The results consistently demonstrated that both the cycle number and the Coulombic efficiency values followed the same performance trend: HPJH-AlSi > HPA-AlSi > As-cast-AlSi > HP-AlSi > Al (**Figure R29**). Specifically, the HP-AlSi sample achieved 100 cycles with an average Coulombic efficiency of 95.69%, while the As-cast-AlSi sample reached 275 cycles with an average Coulombic efficiency of 96.22%. This indicates that the performance of As-cast-AlSi surpassed that of HP-AlSi. This trend was also evident in the symmetric cell performance of the samples. In particular, the HP-AlSi sample demonstrated a symmetric cell cycling time of 25 h, whereas the As-cast-AlSi sample achieved 30 h of cycling (**Figure R30**). Additionally, we further compared the electrochemical performance of these samples using radar plots (**Figure R31**), the specific data of which are also placed in **Table R2**, which consistently confirmed the observed performance trend. These results provide further evidence supporting our conclusions.

Second, regarding the scattered data for the Coulombic efficiency during the cycling process, we believe that this phenomenon can be attributed to the following reasons: **1)** The behavior of Na plating and stripping on the surface of the current collector becomes increasingly unstable as the cycle number increases. Particularly at high cycle numbers, such as beyond 75 cycles in the HP-AlSi system and beyond 175 cycles in the As-cast-AlSi samples, repeated deposition and stripping of Na metal can result in uneven volume expansion and dendrite growth, leading to structural degradation. This degradation causes the gradual accumulation of 'dead Na', which reduces the effective Na reversible capacity, thus causing fluctuations in the Coulombic efficiency [*Angew. Chem. Int. Ed.*, 2023, 62(47), e202312413; *Nano Energy*, 2021, 80, 105563; *Nano Energy*, 2022, 97: 107202; *J. Mater. Chem. A*, 2020, 8(29), 14757-14768]. **2)** During the cycling process, side reactions in the electrolyte lead to the gradual thickening, rupture and regeneration of the SEI film formed by the reaction of metal Na with the electrolyte, resulting in continuous changes in the chemical environment of the electrode surface. Meanwhile, the

rupture and reconstruction of the SEI film will further affect the Na plating and stripping process, increase the interfacial impedance, and thus lead to the fluctuation of the Coulombic efficiency [Chem. Soc. Rev., 2020, 49(12), 3783-3805; Chem. Synth., 2022, 2, 16; Adv. Funct. Mater., 2020, 30(52), 2004891]. In addition, the depletion and passivation of the electrolyte with the increase of the number of cycles further affects the reversibility of the reaction, resulting in the scattered electrochemical behaviour [Energy. Environ. Sci., 2017, 10(9), 1936-1941; Chem. Eng. J., 2024, 492, 152198; ACS Energy Lett. 2017, 2(9), 2051-2057]. In summary, the fluctuations in Coulombic efficiency are primarily caused by multiple factors, including the non-uniformity of Na plating/stripping process, increased interfacial resistance, and the instability of the SEI layer. These phenomena are common and have been widely reported in the literature [Angew. Chem. Int. Ed., 2023, 62(47), e202312413; Sci. Adv., 2022, 8(19), eabm7489; Small, 2021, 17(12), 2007578; ACS Energy Lett. 2017, 2(9), 2051-2057; Adv. Energy Mater., 2020, 10(44), 2002308].

Third, regarding the valuable suggestions on experimental errors and statistical dispersion, we have repeated the experiments for each material and the results have been presented in the following **Figure R32**. Additionally, we have plotted the violin plots based on three repeated experiments for each of the five collectors. The results indicate that the statistical dispersion and experimental error are the largest for Al and the smallest for HPJH-AlSi. Specifically, the ranking is as follows: HPJH-AlSi < HPA-AlSi < As-cast AlSi < HP-AlSi < Al. This trend is in line with other experimental observations in our manuscript, further validating the reproducibility and consistency of our results.

Finally, we have included the plot for the HPJH-AlSi half-cell in the updated version of the manuscript, as shown in the figure below (**Figure R33**). This plot illustrates the stable cyclic performance of the HPJH-AlSi system with an average Coulombic efficiency of 99.71% maintained 4000 cycles, highlighting the enhanced stripping/plating behavior due to the incorporation of nanotwinned nucleation sites, which promote uniform Na deposition and prevent dendritic growth.

We have made the following modifications to the manuscript:

*The Na//HPJH-AlSi cells, employing a current density of 3 mA cm⁻² and a cycling capacity of 1 mAh cm⁻², demonstrate a stable plating/stripping process over 4000 cycles (cumulative capacity of 4 Ah cm⁻²) with an average Coulombic efficiency of 99.71% (**Fig. 5a**). Moreover, the potential curves corresponding to these cycles are stable, as confirmed by a minimal voltage polarization (~43.3 mV), suggesting the robust stability of the precipitated NT-Si anchoring sites (**Fig. 5b**). {Page 14, Line 5}*

Figure R29. Coulombic efficiencies of Na plating/stripping on the collectors at current densities of 3 mA cm^{-2} and areal capacities of 1 mAh cm^{-2} .

Figure R30. Voltage profiles of symmetric cells assembled with HP-AISi and As-cast AISi at 5 mA cm^{-2} and 5 mAh cm^{-2} .

Figure R31. Radar plots of the five main electrochemical properties of different collectors.

Table R2. Summary of electrochemical properties of five types of collectors prepared in our work.

	Average CE value (%)	CE cycle number (n)	CE polarization (mV)	Symmetrical battery cycle time (h)	Lowest battery overpotential (mV)
HPJH-AISi	99.71	4000	43.7	5300	23
HPA-AISi	98.38	1500	44.0	1280	30
As-cast-AISi	96.22	275	48.3	30	46
HP-AISi	95.69	100	80.6	25	51
Al	94.46	40	141.6	24	53

Figure R32. (a-e) Triple Coulombic efficiency test for plating/stripping Na on (a) Al, (b) HP-AISi, (c) As-cast-AISi, (d) HPA-AISi, (e) HPJH-AISi. (f) Violin plots for all Coulombic efficiency tests.

Figure R33. (a) Coulombic efficiencies of Na plating/stripping on the HPJH-AlSi, (b) Voltage-capacity profiles of the HPJH-AlSi at 3 mA cm^{-2} and 1 mAh cm^{-2} .

5) Figure 23- Can the voltage profile be provided for 80% DOD, which is usually the standard in performance metrics to allow for comparisons across different technologies.

Response: We are grateful for your thoughtful comment. We have added the voltage profile for the HPJH-AlSi alloy under the conditions of 80% DOD, with a current density of 8 mA cm^{-2} and a deposition capacity of 8 mAh cm^{-2} . The results indicate that HPJH-AlSi can endure stable cycling for up to 800 h under these challenging conditions (Figure R34). This voltage profile provides a direct basis for comparison across different technologies, demonstrating the superior performance of HPJH-AlSi, especially in terms of cycle life and stability at high discharge depths.

We have made the following modifications to the manuscript:

Even under more stringent conditions— 8 mA cm^{-2} current density, 8 mAh cm^{-2} capacity, and 80 % depth of discharge—the HPJH-AlSi/Na symmetric cell is capable of stable cycling for up to 800 h (Supplementary Fig. 28). {Page 15, Line 1}

Figure R34. Voltage profiles of symmetric cells assembled with HPJH-AlSi at 80 % DOD, 8 mA cm^{-2} and 8 mAh cm^{-2} .

6) How is the total active weight calculated? In sodiated or de-sodiated state?

Response: We are grateful for your thoughtful comment. The total active material weight is calculated based on the mass of the $\text{Na}_3\text{V}_2(\text{PO}_4)_3$ (NVP) loaded on the cathode and the mass of Na deposited on the anode current collector.

Specifically, the NVP on the cathode is a commercially available material that has not undergone further sodiation treatment, and its morphology is shown in the image below (**Figure R35**). In addition, the active mass on the anode is the Na deposited onto the current collector, which corresponds to a deposition capacity of 2 mAh cm^{-2} , and can be considered as sodiated. The morphology of the deposited Na is shown in the provided **Figure R36**.

Furthermore, the calculation of specific values of active mass, as well as the evaluation of energy and power densities for the full cell have been provided later in Supplementary Table 5. Specifically, The evaluation of energy densities (E) and power densities (P) relies on the total mass of the active electrodes, as expressed by the following equations:

$$E = \frac{VCm_{NVP}}{m_{NVP} + m_{Na}}$$

where V represents the average discharge voltage and C denotes the specific capacity, m_{NVP} and m_{Na} correspond to the masses of the NVP cathode and the host Na, respectively.

Power density (P) is determined using the following equation:

$$P = \frac{EI}{C}$$

In this equation, where I represents the specific current density.

Figure R35. (a, b) SEM images of NVP cathode materials and (c-f) corresponding EDS mappings.

Figure R36. (a-c) Top view SEM images of Na deposited on HPJH-AlSi, HPA-AlSi, and Al with the capacities of 2 mAh cm^{-2} . (d-f) Cross-section SEM images of Na deposited on HPJH-AlSi, HPA-AlSi, and Al with the capacities of 2 mAh cm^{-2} .

7) *It would be beneficial to include more detailed analysis and discussion regarding the underlying mechanisms driving the ultrafast sodium deposition observed. Cyclic voltammetry and differential plot (dQ/dV) could be utilized for the same.*

Response: We are grateful for your thoughtful comment. We have conducted additional cyclic voltammetry (CV) and differential capacity (dQ/dV) measurements to further elucidate the underlying mechanisms driving the ultrafast Na deposition dynamics.

First, we assemble half-cells using HPJH-AlSi/Na, HPA-AlSi/Na, and Al/Na and conduct CV tests within a potential window of -0.2 V to 1 V vs. Na/Na^+ at a scanning rate of 5 mV/s . The CV results, as shown in **Figure R37**, demonstrate HPJH-AlSi has the highest peak current, suggesting that the presence of NT-Si contributed to the fastest Na dynamics of HPJH-AlSi [*Energy. Enviro. Sci.*, 2017, 10(9), 1936-1941; *Sci. Adv.*, 2022, 8(44), eabq6321]. Additionally, throughout the Na deposition/stripping processes on all three current collectors, no alloying peaks between Al or Si and Na are observed, confirming the inert nature of Al and Si in these electrochemical reactions.

Next, we integrate the charge-discharge curves of full cells to obtain the dQ/dV plots. The results, as shown in **Figure R38**, reveal oxidation and reduction peaks that align well with the voltage platforms of Na plating and extraction in $\text{Na}_3\text{V}_2(\text{PO}_4)_3$ (NVP) cells. No characteristic peaks related to the alloying reactions of Al or Si with Na are detected. Moreover, the HPJH-AlSi/Na||NVP exhibits the largest peak intensity and the lowest polarization compared to Al/Na||NVP and HPA-AlSi/Na||NVP, indicating that HPJH-AlSi/Na retains the highest active Na content during cycling [*Nano Energy*, 2022, 97, 107203; *Energy Storage Mater.* 2024, 73, 103784]. Additionally, only in the

Al/Na||NVP cell is a reduction peak corresponding to the dissolution of bulk Na⁺ observed, whereas no such peak is detected in the HPJH-AlSi/Na||NVP cell. This indicates the presence of a significant amount of inactive Na in the Al/Na||NVP cell [Nano Energy, 2022, 97, 107203]. In contrast, due to the ultrafast Na dynamics in the HPJH-AlSi, uniform Na deposition and stripping are achieved during cycling, significantly reducing the formation of inactive Na.

We have made the following modifications to the manuscript:

CV tests further demonstrate that HPJH-AlSi has the highest peak current, suggesting that the presence of NT-Si contributed to the fastest Na dynamics in the alloy (Supplementary Fig. 18). Additionally, throughout the Na deposition/stripping processes on all three collectors, no alloying peaks between Al or Si and Na are observed. This confirms the inert nature of Al and Si in these electrochemical reactions, consistent with the voltage profiles during deposition. {Page 10, Line 16}

We have also made the following modifications to the supporting information:

The dQ/dV results reveal oxidation and reduction peaks that align well with the voltage platforms of Na plating and extraction in Na₃V₂(PO₄)₃ (NVP) cells. No characteristic peaks related to the alloying reactions of Al or Si with Na are detected. Moreover, the HPJH-AlSi/Na||NVP exhibits the largest peak intensity and the lowest polarization compared to Al/Na||NVP and HPA-AlSi/Na||NVP, indicating that HPJH-AlSi/Na retains the highest active Na content during cycling. Additionally, only in the Al/Na||NVP cell is a reduction peak corresponding to the dissolution of bulk Na⁺ observed, whereas no such peak is detected in the HPJH-AlSi/Na||NVP cell. This indicates the presence of a significant amount of inactive Na in the Al/Na||NVP cell. In contrast, due to the ultrafast Na dynamics in the HPJH-AlSi, uniform Na deposition and stripping are achieved during cycling, significantly reducing the formation of inactive Na. {Supporting information, Page 39, Line 4}

Figure R37. Cyclic voltammograms for Na deposition/stripping over different collectors at a scan rate of 5 mV s⁻¹.

Figure R38. (a) Typical voltage profiles of different cells at the specific current of 1 C. (b) The corresponding dQ/dV plot obtained from the typical voltage profiles.

8) “The microstructural morphologies and elemental compositions were observed through SEM (FEI Helios G4CX) at an accelerating voltage of 5 kV for SEM imaging and at 20 kV for EDS mapping”- Sodium has a low melting point of 97.80C. In a high-vacuum SEM, the electron beam can cause localized heating of the sample. Since sodium has a low melting point, even modest beam currents or prolonged exposure may cause the sodium to melt or deform. How was the risk of surface bubbling/ melting managed at high voltage like 20 kV? Also, mention the dwell time applied.

Response: Thank you for your valuable comment. To address the concern regarding the potential melting or deformation of Na during SEM and EDS measurements, we implemented several strategies to mitigate this risk.

Presence of SEI Layer: The Na surface is covered by a stable SEI composed of compounds such as NaF and Na₂O. These compounds not only have significantly higher melting points than Na, but their thermal conductivities also contribute to minimizing localized heating. Specifically, the thermal conductivities are as follows: NaF: 0.9 W/m·K; Na₂O: 9.5 W/m·K [Sol. Energy Mater. Sol. Cells, 2014, 126: 11-25; Westinghouse Electric Company, 2015, No. DOE-WEC-0000611-3]. Therefore, the SEI layer not only protects the Na from direct exposure to the electron beam but also acts as a thermal insulator, minimizing localized heating.

Working Distance and High Vacuum Conditions: During SEM and EDS measurements, the working distance between the sample and the electron source was maintained at 8 mm. In addition, high vacuum conditions further aid in heat dissipation, minimizing the chances of localized temperature increases that could lead to Na melting. Under these conditions, the heat transferred to the sample is insufficient to cause Na deformation.

Controlled Dwell Time: We controlled the dwell time during both SEM and EDS measurements. For SEM imaging at 5 kV, the exposure time was limited to approximately 5 s, while for EDS mapping at 20 kV, the exposure time was limited to 2 min. This short

dwelling time effectively reduces the risk of localized heating and Na surface melting. Moreover, the results showed that the EDS mapping was consistent with the morphology presented in the SEM images, and no melting of Na was observed (**Figure R39**).

Experimental Precedents: The parameters we employed, including 5 kV for SEM imaging and 20 kV for EDS mapping, are commonly used in Na metal battery research. Previous studies with similar setups and conditions have not reported any melting or deformation of Na surfaces, supporting the reliability of our approach [*Nano Lett.*, 2017, 17(9), 5862-5868; *Proc. Natl. Acad. Sci.*, 2024, 121(5), e2316914121; *Angew. Chem. Int. Ed.*, 2021, 60(4): 2110-2115; *Small*, 2023, 19(21): 2207638].

Given the thermal insulation provided by the SEI layer, the carefully controlled dwell times, and the high vacuum conditions, the risk of Na melting or surface bubbling was minimized, and no such issues were observed during our tests. Also, regarding the mention of the applied dwell time, we have also added to the manuscript.

We have made the following modifications to the manuscript:

The microstructural morphologies and elemental compositions were observed through SEM (FEI Helios G4CX) at an accelerating voltage of 5 kV for SEM imaging and at 20 kV for EDS mapping. A working distance of 8 mm was maintained during imaging. The dwell time for each SEM image was set to 5 s, while for EDS mapping, the dwell time was 2 min. These settings ensured accurate data collection and minimized the risk of Na melting during the analysis. {Page 20, Line 24}

Figure R39. SEM images and corresponding EDS spectra of (a, c) top view and (b, d) cross-section of Si-coated-Al electrode after cycling.

9) Authors are advised to perform XPS analysis of the HPJH-AlSi/Na, HPA-AlSi/Na and Al/Na electrodes after cycling to understand the chemical nature of oxides formed and correlate it with cell performance.

Response: Thank you for your valuable suggestion. We have conducted XPS analysis on the HPJH-AlSi/Na, HPA-AlSi/Na, and Al/Na electrodes after cycling to investigate the chemical nature of surface oxides and correlate these findings with cell performance (**Figure R40**). The XPS results reveal that under the same analyses [*Adv. Mater.*, 2022, 34(1), 2106005; *Energy Environ. Sci.*, 2024, 17(3), 1216-1228], HPJH-AlSi/Na exhibits the lowest oxide content, HPA-AlSi/Na shows a moderate level of oxides, and Al/Na displays the highest oxide content after cycling. This trend can be attributed to the differences in surface morphology and corresponding side reactions during cycling. The uniform and dense surface morphology of HPJH-AlSi/Na leads to fewer side reactions and, therefore, minimal oxide formation, which aligns with its superior electrochemical performance. In contrast, the Al/Na electrode, with its dendritic surface morphology, shows extensive side reactions, resulting in the highest oxide content, which corresponds to its inferior performance. The intermediate performance of HPA-AlSi/Na is consistent with its relatively uneven surface morphology and moderate oxide content.

We have made the following modifications to the manuscript:

XPS analysis further confirms that the HPJH-AlSi/Na electrode exhibits the lowest oxide formation, which correlates with its minimal parasitic reactions and superior performance (Supplementary Fig. 32). The Al/Na electrode shows the highest oxide content, consistent with its poor performance due to severe surface reactions. In comparison, the HPA-AlSi/Na electrode exhibits a moderate level of oxide formation, corresponding to its performance between HPJH-AlSi and Al/Na. {Page 15, Line 15}

Figure R40. (a-c) XPS patterns of high-resolution O 1s of different electrodes after 10 cycles. (d) The relative ratio of Na vs. O in surface layer of three electrodes according to the XPS data.

10) Fig. 2- SEM micrographs show the images of morphology as a combination of substrate and electrolyte. Fig. 2c shows more granular, uniform and compact morphology than the one used for deposition (Fig. 2b). In this regard, why 1M NaClO₄ in DEC:EC as electrolyte was not preferred?

Response: Thank you for your detailed comments regarding the selection of electrolytes for further electrochemical testing. The decision to select 1 M NaPF₆ in diglyme =100 vol% as the electrolyte for the subsequent electrochemical tests, despite the seemingly more granular, uniform, and compact morphology observed in 1 M NaClO₄ in DEC:EC=1:1 vol% with 5% FEC (as shown in Fig. 2c), was made based on the following considerations:

1. Stability and Compatibility: While 1 M NaClO₄ in DEC:EC=1:1 vol% with 5% FEC exhibits a more uniform Na deposition in SEM images, its electrochemical stability over extended cycling was found to be less optimal compared to 1 M NaPF₆ in diglyme =100 vol%. In previous reports, the 1 M NaClO₄ in DEC:EC=1:1 vol% with 5% FEC electrolyte demonstrated significant capacity fading and lower Coulombic efficiency, particularly under higher current densities [*Angew. Chem. Int. Ed.*, 2023, 62(6): e202214372]. In addition, due to the reactivity of esters with Na metal, a less stable SEI is often formed [*Energy Storage Mater.*, 2024, 67, 103211]. This results in the formation of a thicker and more resistive SEI, which degrades battery performance. On the other hand, we have shown in our previous work that 1 M NaPF₆ in diglyme =100 vol% displayed superior stability in maintaining the integrity of the Na electrode, which is crucial for the prolonged operation of Na metal batteries [*ACS Central Sci.*, 2015, 1(8), 449-455].

2. Previous Research and Comparison: 1 M NaPF₆ in diglyme =100 vol% has been widely studied and reported in the literature for Na metal batteries due to its well-established compatibility with Na anodes [*Adv. Mater.*, 2024, 36(15), 2310347; *Adv. Funct. Mater.*, 2024, 34(21), 2314954; *Sci. Adv.*, 2022, 8(19), eabm7489; *Adv. Mater.*, 2023, 35(32), 2301967; *Sci. Adv.*, 2023, 9, eadh8060; *Adv. Funct. Mater.*, 2019, 29(3), 1805946; *Nat. Commun.*, 2021, 12(1), 5786]. Its use allows for more straightforward comparisons with other studies, making the results of our work more relevant to the broader scientific community. Although 1 M NaClO₄ in DEC:EC=1:1 vol% with 5% FEC electrolyte is promising, their electrochemical performance still lags behind in terms of long-term stability and efficiency.

In conclusion, while the 1 M NaClO₄ in DEC:EC=1:1 vol% with 5% FEC electrolyte exhibits some advantages in terms of initial Na deposition morphology, the overall electrochemical stability, and long-term performance of 1 M NaPF₆ in diglyme =100 vol% made it the more suitable choice for our in-depth electrochemical testing.

11) Fig. 22 c (supplementary material) shows stripping/plating stability of HP-AlSi system. The plot shows abrupt behavior at 100th cycle. Please explain it. Also, provide the plot for HPJH-AlSi half cell.

Response: Thank you for your insightful comments. The abrupt behavior in overpotential at the 100th cycle during the stripping/plating stability test in the HP-AlSi system can be attributed as follows:

Specifically, the high-pressure solid solution treatment results in the absence of Si solute on the HP-AlSi alloy surface, as confirmed by XRD analysis (**Figure R41**). Consequently, the ability of HP-AlSi to enhance Na deposition dynamics is limited, leading to uneven Na deposition. The occurrence of such uneven deposition will cause a large overpotential for deposition/stripping and sharp fluctuations in Coulombic efficiency values (**Figure R42**). More seriously, repeated Na plating/stripping as well as volume changes over a long period of cycling may lead to the accumulation of inactive Na ("dead Na") [*Adv. Energy Mater.*, 2022, 12(43), 2202293; *ACS nano*, 2022, 16(10), 17197-17209; *Adv. Energy Mater.*, 2022, 12(32), 2200990]. Moreover, localized mechanical stress at the Na/HP-AlSi interface could cause partial detachment of the deposited Na, further increasing interfacial resistance [*Nano Energy*, 2022, 97, 107203; *J. Mater. Chem. A*, 2020, 8(29), 14757-14768]. These combined effects contribute to the observed rise in overpotential after extended cycling.

In addition, this phenomenon of stripping/plating changes during the cycle of unmodified materials is common and has been reported in many previous articles [*Energy Environ. Sci.*, 2021, 14(12), 6381-6393; *Nat. Commun.*, 2021, 12(1), 5786; *Nano Lett.*, 2017, 17(9), 5862-5868; *Nano Lett.*, 2020, 20(8), 6112-6119], which is consistent with the results of our tests.

In response to the second part of the comment, we have included the plot for the HPJH-AlSi half-cell in the updated version of the manuscript, as shown in the figure below (**Figure R43**). This plot illustrates the stable cyclic performance of the HPJH-AlSi system with an average Coulombic efficiency of 99.71% maintained 4000 cycles, highlighting the enhanced stripping/plating behavior due to the incorporation of nanotwinned nucleation sites, which promote uniform Na deposition and prevent dendritic growth.

We have made the following modifications to the manuscript:

*The Na||HPJH-AlSi cells, employing a current density of 3 mA cm^{-2} and a cycling capacity of 1 mAh cm^{-2} , demonstrate a stable plating/stripping process over 4000 cycles (cumulative capacity of 4 Ah cm^{-2}) with an average Coulombic efficiency of 99.71% (**Fig. 5a**). Moreover, the potential curves corresponding to these cycles are stable, as confirmed by a minimal voltage polarization ($\sim 43.3 \text{ mV}$), suggesting the robust stability of the precipitated NT-Si anchoring sites (**Fig. 5b**). {Page 14, Line 5}*

Figure R41. XRD patterns of the different Al-Si alloys.

Figure R42. (a) Coulombic efficiencies of Na plating/stripping on the HP-AlSi, (b) Voltage-capacity profiles of the HP-AlSi at 3 mA cm^{-2} and 1 mAh cm^{-2} .

Figure R43. (a) Coulombic efficiencies of Na plating/stripping on the HPJH-AlSi, (b) Voltage-capacity profiles of the HPJH-AlSi at 3 mA cm^{-2} and 1 mAh cm^{-2} .

12) The manuscript should not start with a symbol like “Na”. Authors are advised to use name instead of symbol in the beginning. Also, a “conclusion section” at the end would be beneficial for readers.

Response: Thank you for your insightful comments. In response to your suggestion on the beginning of the manuscript, we have revised the manuscript to avoid starting with a symbol like “Na.” Instead, we now begin with the full name, “Sodium,” to ensure clarity and adherence to academic writing standards.

We have made the following modifications to the manuscript:

Sodium (Na) metal batteries are considered promising solutions for next-generation electrochemical energy storage because of their low costs and high energy densities.....
{Page 2, Line 1}

Regarding the second point, we have added a “Conclusion” section at the end of the manuscript, as suggested. The conclusion is provided as follows:

Conclusion

In summary, differing from traditional methods, we have designed a nanotwinned nucleation site alloy collector using the HPJH method to enhance the Na deposition dynamics. This approach transforms Na deposition from diffusion-controlled process to reaction-controlled process, facilitating spherical Na deposition and dendrite-free growth. As a result, the cells equipped with HPJH-AlSi have achieved high capacities and an ultralong cycling lifespans under demanding conditions, such as high utilization and large-rate circulation. This nanotwinned alloy strategy is equally scalable to other alloy materials, providing a significant impetus for the advancement of of dendrite-free metal batteries. {Page 19, Line 3}

REVIEWERS' COMMENTS

Reviewer #1 (Remarks to the Author):

In view that the authors have addressed my concerns, I recommend the present manuscript for publication in Nature Communications.

Reviewer #2 (Remarks to the Author):

I am satisfied with the changes made. And there are some minor points additionally needed to be clarified, but without further peer evaluation.

1. (In Fig.3a, the current density used for Na deposition needs to be provided. How about using the high current density (i.e., 5, 7,10 mA cm⁻²) for Na nucleation and growth with spherical deposition
2. the parameters of the formula in Supplementary Note 1 do not correspond to the parameters in the Supplementary Table 1.

Reviewer Comments:

I have thoroughly reviewed the revised manuscript titled "A Nanotwinned-Alloy Strategy Enables Ultrafast Sodium Deposition Dynamics" and would like to commend the authors for addressing each of the questions and providing comprehensive answers, backed with clear presentation of results, logically-structured arguments and appropriate analysis that supports the conclusions drawn. The writing is coherent and the arguments are logically structured. I appreciate the attention-to-detail and the clear explanations of complex concepts, which demonstrate a deep understanding of the dendrite-free metal batteries.

The study is both well-conceived and well-executed. The authors have successfully addressed the research question and have contributed novel findings that will be of great interest to the readership.

I am confident that this paper will make a valuable addition to the journal and I fully support its publication.

Recommendation: Accept with no further revisions.

Reply to the report of referee #1

Comment of Referee #1:

Reviewer 1

In view that the authors have addressed my concerns, I recommend the present manuscript for publication in Nature Communications.

Response: We appreciate the reviewer's recommendation for publishing our manuscript on Nature Communications.

Reply to the report of referee #2

Comment of Referee #2:

Reviewer 2

I am satisfied with the changes made. And there are some minor points additionally needed to be clarified, but without further peer evaluation.

1) In Fig.3a, the current density used for Na deposition needs to be provided. How about using the high current density (i.e., 5, 7,10 mA cm⁻²) for Na nucleation and growth with spherical deposition.

Response: We are grateful for your thoughtful comment. The *in-situ* Na deposition process in nano-battery was conducted at a current density of 1 mA cm⁻² which is consistent with the current density used in Figure 2 from the main manuscript and controlled by changing the bias voltage from 0 V to -5 V. Moreover, in our evaluation of *in-situ* electron microscopy, we acknowledge that it is a technique that is both time-intensive and costly. To more accurately capture the deposition morphology of the battery under practical conditions, we opted to conduct tests using coin cells at higher current densities, rather than performing *in-situ* electron microscopy at even greater current densities (the max value of 30 mA cm⁻² in our lab). The SEM results, as depicted in Fig. R1, reveal that sodium deposition on the HPJH-AlSi current collector forms spherical structures at current densities of 5, 7, and even 10 mA cm⁻². This indicates that the HPJH-AlSi current collector maintains spherical nucleation and growth even at elevated current densities.

Figure R1. Na deposition morphology on the surface of HPJH-AlSi collector at (a) 5 mA cm⁻², (b) 7 mA cm⁻² and (c) 10 mA cm⁻². The deposition capacity, 0.1 mAh cm⁻².

We have made the following changes in the revised manuscript:

After applying a negative bias (0 V to -5 V) and conducting at a current density of 3 mA cm⁻² to the HPJH-AlSi, HPA-AlSi or Al against the Na electrode, the electrochemical deposition process was initiated. {Page 23, Line 8}

2) *The parameters of the formula in Supplementary Note 1 do not correspond to the parameters in the Supplementary Table 1.*

Response: We thank the reviewer for pointing out the inconsistency between the parameters in Supplementary Note 1 and those in Supplementary Table 1. After careful review, we have identified the source of this discrepancy and revised both the supplementary note and the table to ensure consistency and clarity. The updated content is as follows:

Supplementary Note 5

The deposition morphology on a substrate is determined by the relative magnitudes of the electrochemical reaction rate and the surface diffusion rate. For Na deposition on various substrates, these parameters are related to the experimental current density and the self-diffusion coefficient of Na. The electrochemical reaction rate, k_e , can be derived from the experimental current density, J , as follows:

$$k_e = \frac{Ja^2}{F} N_A \quad (2)$$

Where a is the size of one lattice cell, F is Faraday's constant, and N_A is Avogadro's constant. The values of these parameters are listed in Supplementary Table 1. This paper introduces three primary current densities: 1, 3 and 5 mA cm⁻², respectively. Correspondingly, the k_e values are 22.47, 67.41 and 112.34 s⁻¹, respectively.

The surface diffusion rate, k_d , is calculated using Equation S4 and is dependent on the activation energy barrier for self-diffusion, E_{diff} , temperature, T , and jump frequency for Na diffusion, v . The literature reports average values of jump frequency, which range between ca. 10¹²-10¹³ s⁻¹. Therefore, a value of $v=5 \times 10^{12}$ s⁻¹ has been utilized where k_B represents the Boltzmann constant.

$$k_d = v \exp\left(\frac{-E_{diff}}{k_B T}\right) \quad (3)$$

The self-diffusion energy barrier, E_{diff} , determines the magnitude of the diffusion rate and therefore must be accurately measured to capture the underlying dynamics of the Na system. In this paper, we used DFT calculations to determine the surface Na diffusion barriers of NT-Si, Si, and Al, which are 0.86, 0.62, and 0.48 eV, respectively. To simplify the calculations, we consider the corresponding diffusion rates for the three types of collectors (Al, HPA-AlSi and HPJH-AlSi) to be 1.38×10^5 , 1.92×10^2 , and 1.78×10^{-2} s⁻¹. {Supporting information Page 50}

Supplementary Table 1. Model parameters.

Parameters		Values	Units
a	Lattice cell dimension	0.6	nm
J	Average current density	1,3,5	mA cm ⁻²
F	Faraday's constant	96487	C/mol
N _A	Avogadro's constant	6.022×10 ²³	L/mol
T	Operating temperature	300	K
v	Vibration frequency	5×10 ¹²	s ⁻¹
k _B	Boltzmann's constant	8.617×10 ⁻⁵	eV K ⁻¹

{Supporting information Page 55}

Reply to the report of referee #3

Comment of Referee #3:

Reviewer 3

I have thoroughly reviewed the revised manuscript titled "A Nanotwinned-Alloy Strategy Enables Ultrafast Sodium Deposition Dynamics" and would like to commend the authors for addressing each of the questions and providing comprehensive answers, backed with clear presentation of results, logically-structured arguments and appropriate analysis that supports the conclusions drawn. The writing is coherent and the arguments are logically structured. I appreciate the attention-to-detail and the clear explanations of complex concepts, which demonstrate a deep understanding of the dendrite-free metal batteries.

The study is both well-conceived and well-executed. The authors have successfully addressed the research question and have contributed novel findings that will be of great interest to the readership.

I am confident that this paper will make a valuable addition to the journal and I fully support its publication.

Recommendation: Accept with no further revisions.

Response: We appreciate the reviewer's positive comments on our manuscript and recommendation for publication.